# Toward Student-oriented Teacher Network Training for Knowledge Distillation

**Chengyu Dong**[1]   **Liyuan Liu**[2]   **Jingbo Shang**[1]
[1]University of California, San Diego   [2]Microsoft Research
`{cdong, jshang}@ucsd.edu  lucliu@microsoft.com`

## Abstract

How to conduct teacher training for knowledge distillation is still an open problem. It has been widely observed that a best-performing teacher does not necessarily yield the best-performing student, suggesting a fundamental discrepancy between the current teacher training practice and the ideal teacher training strategy. To fill this gap, we explore the feasibility of training a teacher that is oriented toward student performance with empirical risk minimization (ERM). Our analyses are inspired by the recent findings that the effectiveness of knowledge distillation hinges on the teacher's capability to approximate the true label distribution of training inputs. We theoretically establish that ERM minimizer can approximate the true label distribution of training data as long as the feature extractor of the learner network is Lipschitz continuous and is robust to feature transformations. In light of our theory, we propose a teacher training method *SoTeacher* which incorporates Lipschitz regularization and consistency regularization into ERM. Experiments on benchmark datasets using various knowledge distillation algorithms and teacher-student pairs confirm that *SoTeacher* can improve student accuracy consistently.

## 1 Introduction

Knowledge distillation aims to train a small yet effective *student* neural network following the guidance of a large *teacher* neural network (Hinton et al., 2015). It dates back to the pioneering idea of model compression (Buciluǎ et al., 2006) and has a wide spectrum of real-world applications, such as recommender systems (Tang & Wang, 2018; Zhang et al., 2020), question answering systems (Yang et al., 2020; Wang et al., 2020) and machine translation (Liu et al., 2020).

Despite the prosperous interests in knowledge distillation, one of its crucial components, teacher training, is largely neglected. The existing practice of teacher training is often directly targeted at maximizing the performance of the teacher, which does not necessarily transfer to the performance of the student. Empirical evidence shows that a teacher trained toward convergence will yield an inferior student (Cho & Hariharan, 2019) and regularization methods benefitting the teacher may contradictorily degrade student performance (Müller et al., 2019). As also shown in Figure 1, the teacher trained toward convergence will consistently reduce the performance of the student after a certain point. This suggests a fundamental discrepancy between the common practice in teacher training and the ideal learning objective of the teacher that orients toward student performance.

In this work, we explore both the theoretical feasibility and practical methodology of training the teacher toward student performance. Our analyses are built upon the recent understanding of knowledge distillation from a statistical perspective. In specific, Menon et al. (2021) show that the soft prediction provided by the teacher is essentially an approximation to the true label distribution, and true label distribution as supervision for the student improves the generalization bound compared to one-hot labels. Dao et al. (2021) show that the accuracy of the student is directly bounded by the distance between teacher's prediction and the true label distribution through the Rademacher analysis.

Based on the above understanding, a teacher benefitting the student should be able to learn the true label distribution of the *distillation data* [1]. Since practically the distillation data is often reused from

---

[1]For simplicity, we refer to the training data of the student model in knowledge distillation as the distillation data (Stanton et al., 2021)

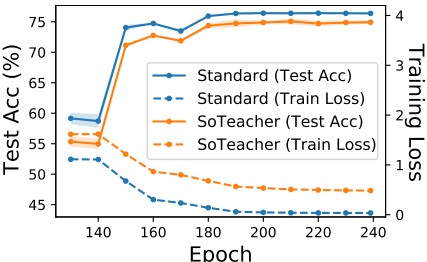
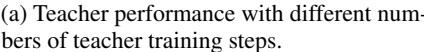
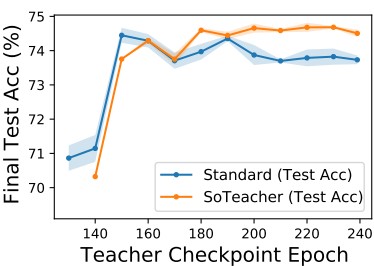

(a) Teacher performance with different numbers of teacher training steps.

(b) Student performance when distilled from different teacher checkpoints.

Figure 1: We train teacher models on CIFAR-100, saving a checkpoint every 10 epochs, and then use this checkpoint to train a student model through knowledge distillation. With our method, the teacher is trained with a focus on improving student performance, leading to better student performance even if the teacher's own performance is not as high.

the teacher's training data, the teacher will have to learn the true label distribution of its own training data. This might appear to be infeasible using standard empirical risk minimization, as the teacher network often has enough capacity to fit all one-hot training labels, in which case, distilling from teacher predictions should largely degrade to directly training with one-hot labels. Existing works tend to evade this dilemma by distilling from teacher predictions only on data that is not used in teacher training (Menon et al., 2021; Dao et al., 2021).

Instead, we directly prove the feasibility of training the teacher to learn the true label distribution of its training data. We show that the standard empirical risk minimizer can approach the true label distribution of training data under a mixed-feature data distribution, as long as the feature extractor of the learner network is Lipschitz continuous and is robust to feature transformations.

In light of our theory, we show that explicitly imposing the Lipschitz and consistency constraint in teacher training can facilitate the learning of the true label distribution and thus improve the student performance. We conduct extensive experiments on two benchmark datasets using various knowledge distillation algorithms and different teacher-student architecture pairs. The results confirm that our method can improve student performance consistently and significantly.

We believe our work is among the first attempts to explore the theory and practice of training a *student-oriented* teacher in knowledge distillation. To summarize, our main contributions are as follows.

- We show that it is theoretically feasible to train the teacher to learn the true label distribution of the distillation data even with data reuse, explaining the effectiveness of the current knowledge distillation practice.
- We show that by adding Lipschitz and consistency regularization during teacher training, it can better learn the true label distribution and improve knowledge consistently.

## 2 PRELIMINARIES

We study knowledge distillation in the context of multi-class classification. Specifically, we are given a set of training samples $\mathcal{D} = \{(x^{(i)}, y^{(i)})\}_{i \in [N]}$, where $[N] := \{1, 2, \cdots, N\}$. $\mathcal{D}$ is drawn from a probability distribution $p_{X,Y}$ that is defined jointly over input space $\mathcal{X}$ and label space $\mathcal{Y} = [K]$. For convenience, we denote $1(y) \in \mathbb{R}^K$ as the one-hot encoding of label $y$.

**Learning Objective of Teacher in Practice.** In common practice, the teacher network $f$ is trained to minimize the empirical risk given a loss function $\ell$, namely

$$\min_f \mathbb{E}_{(x,y) \in \mathcal{D}} \, \ell(f(x), y), \tag{1}$$

where $\mathbb{E}_{(x,y) \in \mathcal{D}}$ is the empirical expectation.

**Ideal Learning Objective of Teacher.** Recent advances in understanding knowledge distillation suggest a teacher should approximate the true label distribution of the distillation data, which is often

reused from the teacher's training data. From this perspective, ideally the teacher should learn the true label distribution of its training data, namely

$$\min_f \mathbb{E}_{(x,y)\in\mathcal{D}} \|f(x) - p^*(x)\|. \tag{2}$$

Here, $p^*(x) := p_{Y|X}(\cdot|x)$ denotes the *true label distribution* of an input $x$, namely the (unknown) category distribution that its label is sampled from, which is not necessarily one-hot. And $\|\cdot\|$ can be an arbitrary $p$-norm.

**Our Research Questions.** One can find that there is a fundamental discrepancy between the learning objective of the teacher in the common practice and the ideal learning objective of the teacher. In particular, minimization of Eq. (1) would lead to $f(x) = 1(y)$ for any input $x$, which significantly deviates from $p^*(x)$ and thus challenges the effectiveness of knowledge distillation. Therefore, in this work, we explore the following two questions.

(i) *Can a teacher network learn the true label distribution of the training data with the standard teacher training practice?*
(ii) *How to train a teacher to better learn the true label distribution and improve student performance?*

We will present our findings of these two questions in Sec. 3 and Sec. 4, respectively.

# 3 THEORETICAL FEASIBILITY TO LEARN TRUE LABEL DISTRIBUTION OF TRAINING DATA

We now explore the theoretical feasibility of training a teacher network that learns the true label distribution of the training data under the empirical risk minimization framework. Unless specifically stated otherwise, by "data" we refer to the data used to train the teacher, and by "network" we refer to the teacher network. Note that throughout the discussion here, our major focus is the *existence* of a proper minimizer, instead of the details of the optimization process.

## 3.1 NOTATIONS AND PROBLEM SETUP

**Notation.** Here, we introduce our notations in addition to the ones mentioned in Section 2. We will use calligraphic typefaces to denote sets, *e.g.*, the dataset $\mathcal{D}$. We use $|\mathcal{D}|$ to denote the size of the set. We use $\circ$ to denote function composition. We use $\tilde{O}(\eta)$ to denote polynomial terms of $\eta$.

**Data Distribution.** We consider a data distribution which we refer to as the *mixed-feature distribution*. Our distribution can be viewed as a simplified version of the "multi-view distribution" introduced in Allen-Zhu & Li (2020). Our distribution can also be viewed as a variation of Latent Dirichlet Allocation (LDA) (Blei et al., 2001), a generative data distribution widely used to model text data.

**Definition 3.1** (Mixed-feature distribution). *We first generate the input $x$ following LDA. We define a feature vocabulary $\mathcal{Z}$ that denotes the names of all possible features in the data inputs. For example, in image classification we can have $\mathcal{Z} = \{$'eye', 'tail', 'wheel', ...$\}$. Now for each data example $i$,*

1. *Sample $M$ feature names from $\mathcal{Z}$, namely $z_m \sim p_Z$, where $p_Z$ is a discrete distribution defined over $\mathcal{Z}$.*

2. *For each feature name $z$, we sample its input representation $x_z \in \mathbb{R}^b$, namely $x_z \sim p_X(\cdot|z)$, where $p_X(\cdot|z)$ is a continuous distribution with finite variance, namely $Var[X|z] \le \nu_z$. The finite variance here means that the representation of the same feature sampled in different inputs should be similar.*

3. *For each feature name $z$, we transform its input representation by a function $\gamma : \mathbb{R}^b \to \mathbb{R}^b$, where $\gamma$ is sampled from a set of possible transformation functions $\mathcal{T}$, namely $x_z := \gamma(x_z)$. For example, in image classification $\gamma$ can be rotation, mirroring, or resizing.*

4. *We concatenate the transformed representations of these features to obtain the input, namely $x = (x_{z_1}, x_{z_2}, \cdots, x_{z_M})$, where $x \in \mathbb{R}^{b \times M}$. This views each data input as a concatenation of $M$ patches and each patch is a $b$-dimensional vector, which follows the assumption in Allen-Zhu & Li (2020).*

*Next, we generate the label $y$. We assume each feature name defines a label distribution $p_Y(\cdot|z)$. The label distribution of an input $x$ is the geometric average of the label distributions of the feature names, namely $y \sim p_Y(\cdot|x) := (\prod_m p_Y(\cdot|z_m))^{1/M}$.*

Note that in our data distribution, the assumption that each patch is a vector is solely for the simplicity of illustration and can be trivially generalized, as our theory is not dependent on it. The specific shape of each patch can be arbitrary, for example, can be a 3-rank tensor (height, width, channel) in image classification.

One difference between our data distribution and "multi-view distribution" is that one can define different label sets $\mathcal{Y}$ for the same input data under our data distribution. This is more realistic as it models datasets that have coarse-grained/fine-grained label sets respectively but the same input data.

**Network Architecture.** We consider a multi-layer neural network that produces probabilistic outputs, namely $f : \mathbb{R}^{b \times M} \to [0,1]^K$. The network consists of a feature extractor and a classification head, namely $f := f_C \circ f_E$, defined as follows respectively.

- $f_E : \mathbb{R}^{b \times M} \to \mathbb{R}^{M \times d}$ denotes the feature extractor, whose architecture can be arbitrary as long as it processes each patch independently. With a little abuse of notation, we will also write $h_m := f_E(x_m)$.
- $f_C : \mathbb{R}^{M \times d} \to \mathbb{R}^K$ denotes the probabilistic classification head, which consists of a 1x1 convolutional layer and a modified Softmax layer, namely $f_C(h) = \widetilde{\text{Softmax}}(w_C h)$, where $w_C \in \mathbb{R}^{1 \times K \times d}$. The 1x1 convolutional layer is similar to the assumption in (Allen-Zhu & Li, 2020), while the modified Softmax is slightly different from the standard Softmax in terms of the denominator, namely $\widetilde{\text{Softmax}}(\hat{h}) = \frac{\exp(1/M \sum_m \hat{h}_m)}{(\prod_m \sum_k \exp(\hat{h}_{m,k}))^{1/M}}$, where $\hat{h} := w_c h$.

**Learning.** Given a dataset $\mathcal{D}$, we train the network under the empirical risk minimization framework as indicated by Eq. (1). We consider cross-entropy loss $\ell(f(x), y) = -1(y) \cdot \log f(x)$.

## 3.2 A Hypothetical Case: Invariant Feature Extractor

For starters, we investigate a hypothetical case where the feature extractor is *invariant*, which means that it can always produce the same feature map given the same feature patch, regardless of which input this feature patch is located at or which transformation is applied to this feature patch. Given such an assumption, showing the learning of true label distribution of the training data can be greatly simplified. We will thus briefly walk through the steps towards this goal to shed some intuitions.

**Definition 3.2** (Invariant feature extractor). *We call $f_E$ an invariant feature extractor, if for any two inputs $i$ and $j$, for any two transformations $\gamma$ and $\gamma'$, $f_E(\gamma(x_m^{(i)})) = f_E(\gamma'(x_{m'}^{(j)}))$, as long as $z_m = z_{m'}$, namely the feature names of the patches $m$ and $m'$ match.*

We first show that given an invariant feature extractor, the minimizer of the empirical risk has the property that its probabilistic prediction of each feature converges to the sample mean of the labels whose corresponding inputs contain this feature.

**Lemma 3.3** (Convergence of the probabilistic predictions of features). *Let $\bar{y}_z := \frac{1}{N} \sum_{\{i|z \in \mathcal{Z}^{(i)}\}} 1(y^{(i)})$, where $\mathcal{Z}^{(i)}$ denote the set of feature names in the $i$-th input, and thus $\{i|z \in \mathcal{Z}^{(i)}\}$ denotes the set of inputs that contain feature $z$. Let $f^*$ be a minimizer of the empirical risk (Eq. (1)) and assume $f_E^*$ is an invariant feature extractor. Let $p_{f^*}(x_z) := Softmax(w_C f_E^*(x_z))$ be the probabilistic prediction of feature $z$. We have*

$$p_{f^*}(x_z) = \bar{y}_z. \tag{3}$$

The intuition here is that since the feature extractor is invariant and the number of possible features is limited, the feature maps fed to the classification head would be a concatenation of $M$ vectors selected from a fixed set of $|\mathcal{Z}|$ candidate vectors, where each vector corresponds to one feature in the vocabulary. Therefore, the empirical risk can be regrouped as $-\frac{1}{N} \sum_i 1(y^{(i)}) \cdot \log f^*(x^{(i)}) = -\frac{1}{M} \sum_{z \in \mathcal{Z}} \bar{y}_z \cdot \log p_{f^*}(x_z)$. Then by Gibbs' inequality, the empirical risk can be minimized only when the probabilistic prediction of each feature $p_{f^*}(x_z)$ is equal to $\bar{y}_z$.

We now proceed to show that the above sample mean of multiple labels $\bar{y}_z$ will converge to the average distribution of these labels, even though they are non-identically distributed. Since each label is a categorical random variable, this directly follows the property of multinomial distributions (Lin et al., 2022).

**Lemma 3.4** (Convergence of the sample mean of labels). *Let $\bar{p}(\cdot|z) = \frac{1}{N} \sum_{\{i|z \in \mathcal{Z}^{(i)}\}} p_Y(\cdot|x^{(i)})$, we have with probability at least $1 - \delta$,*

$$\|\bar{y}_z - \bar{p}(\cdot|z)\| \leq \tilde{O}\left(\sqrt{KN^{-1}M|\mathcal{Z}|^{-1}\delta^{-1}}\right). \tag{4}$$

It is also feasible to achieve the above lemma by applying Lindeberg's central limit theorem, with a weak assumption that the variance of each label is finite.

Next, we show that the average label distribution $\bar{p}(\cdot|z)$ approximates the true label distribution of the corresponding feature $z$.

**Lemma 3.5** (Approximation of the true label distribution of each feature).

$$\|\bar{p}(\cdot|z) - p_Y(\cdot|z)\| \leq \tilde{O}\left(M|\mathcal{Z}|^{-1}\right). \tag{5}$$

The intuition is that since the average distributions $\bar{p}(\cdot|z)$ here are all contributed by the true label distribution of the inputs that contain feature $z$, it will be dominated by the label distribution of feature $z$, given some minor assumption on the sampling process of features when generating each input.

Finally, combining Lemmas 3.3, 3.4, 3.5, one can see that the probabilistic prediction of each feature given by the empirical risk minimizer will approximate the true label distribution of that feature. Subsequently, we can show that the probabilistic prediction of each input approximates the true label distribution of that input, which leads to the following main theorem.

**Theorem 3.6** (Approximation error under a hypothetical case). *Given the setup introduced in Section 3.1, let $f^*$ be a minimizer of the empirical risk (Eq. (1)) and assume $f_E^*$ is an invariant feature extractor. Then for any input $x \in \mathcal{D}$, with probability at least $1 - \delta$,*

$$\|f^*(x) - p^*(x)\| \leq \tilde{O}\left(\sqrt{KN^{-1}M|\mathcal{Z}|^{-1}\delta^{-1}}\right) + \tilde{O}\left(M|\mathcal{Z}|^{-1}\right). \tag{6}$$

### 3.3 REALISTIC CASE

We now show that in a realistic case where the feature extractor is not exactly invariant, it is still possible to approximate the true label distribution, as long as the feature extractor is robust to the variation of features across inputs and also robust to feature transformations.

**Definition 3.7** (Lipschitz-continuous feature extractor). *We call $f_E$ a $L_X$-Lipschitz-continuous feature extractor, if for any two inputs $i$ and $j$, $\|f_E(x_m^{(i)}) - f_E(x_{m'}^{(j)})\| \leq L_X\|x_m^{(i)} - x_{m'}^{(j)}\|$, as long as $z_m = z_{m'}$, namely the feature names of the patches $m$ in $i$ and patch $m'$ in $j$ match.*

**Definition 3.8** (Transformation-robust feature extractor). *We call $f_E$ a $L_\Gamma$-transformation-robust feature extractor, if for any patch $m$ in any input $i$, and for any transformation $\gamma$, $\|f_E(\gamma(x_m^{(i)})) - f_E(x_m^{(i)})\| \leq L_\Gamma$.*

Similar to Lemma 3.3, given a Lipschitz-continuous and transformation-robust feature extractor, the probabilistic predictions of features will still converge to the sample mean of the labels where the corresponding inputs contain this feature, up to some constant error. This requires the assumption that the input representation of the same feature is similar across different inputs, which is intuitive and covered by Definition 3.1. All other lemmas introduced in the hypothetical case still hold trivially as they are not dependent on the network. Therefore, we can have the following result.

**Theorem 3.9** (Approximation error under a realistic case). *Given the setup introduced in Section 3.1, let $f^*$ be a minimizer of the empirical risk (Eq. (1)) and assume $f_E^*$ is a $L_X$-Lipschitz-continuous and $L_\Gamma$-transformation-robust feature extractor. Let $\nu = \max_z \nu_z$. Then for any input $x \in \mathcal{D}$, with probability at least $1 - \delta$,*

$$\|f^*(x) - p^*(x)\| \leq \tilde{O}\left(\sqrt{KN^{-1}M|\mathcal{Z}|^{-1}\delta^{-1}}\right) + \tilde{O}\left(M|\mathcal{Z}|^{-1}\right) + \tilde{O}(L_X\tilde{\delta}^{-0.5}\nu) + \tilde{O}(L_\Gamma). \tag{7}$$

## 4    *SoTeacher*

Based on our theoretical findings, we investigate practical techniques for training the teacher model to more accurately approximate the true label distribution of the training data.

**Lipschitz regularization.**    Theorem 3.9 suggests that it is necessary to enforce the feature extractor to be Lipschitz continuous. This may also be achieved by current teacher training practice, as neural networks are often implicitly Lipschitz bounded (Bartlett et al., 2017). Nevertheless, we observe that explicit Lipschitz regularization (LR) can still help in multiple experiments (See Sect. 5). Therefore, we propose to incorporate a global Lipschitz constraint into teacher's training. For the implementation details, we follow the existing practice of Lipschitz regularization (Yoshida & Miyato, 2017; Miyato et al., 2018) and defer them to Appendix C.

**Consistency regularization.**    Theorem 3.9 also shows that to learn the true label distribution, it is necessary to ensure the feature extractor is robust to feature transformations. This is aligned with the standard practice which employs data augmentation as regularization. However, when data is scarce, it is better to explicitly enforce the model to be robust to transformations. This is known as consistency regularization (CR) (Laine & Aila, 2017; Xie et al., 2020; Berthelot et al., 2019; Sohn et al., 2020) that is widely used in semi-supervised learning.

Considering the training efficiency of consistency regularization, we utilize temporal ensembling (Laine & Aila, 2017), which penalizes the difference between the current prediction and the aggregation of previous predictions for each training input. In this way, the consistency under data augmentation is implicitly regularized since the augmentation is randomly sampled in each epoch. And it is also efficient as no extra model evaluation is required for an input.

To scale the consistency loss, a loss weight is often adjusted based on a Gaussian ramp-up curve in previous works (Laine & Aila, 2017). However, the specific parameterization of such a Gaussian curve varies greatly across different implementations, where more than one additional hyperparameters have to be set up and tuned heuristically. Here to avoid tedious hyperparameter tuning we simply linearly interpolate the weight from 0 to its maximum value, namely $\lambda_{\mathrm{CR}}(t) = \frac{t}{T}\lambda_{\mathrm{CR}}^{\max}$, where $T$ is the total number of epochs.

**Summary.**    To recap, our teacher training method introduces two additional regularization terms. The loss function can thus be defined as $\ell = \ell_{\mathrm{Stand.}} + \lambda_{\mathrm{LR}}\ell_{\mathrm{LR}} + \lambda_{\mathrm{CR}}\ell_{\mathrm{CR}}$, where $\ell_{\mathrm{Stand.}}$ is the standard empirical risk defined in Eq.(1) and $\lambda_{\mathrm{LR}}$ is the weight for Lipschitz regularization. Our method is simple to implement and incurs only minimal computation overhead (see Section 5.2).

## 5    EXPERIMENTS

In this section, we evaluate the effectiveness of our teacher training method in knowledge distillation. We focus on compressing a large network to a smaller one where the student is trained on the same data set as the teacher.

### 5.1    EXPERIMENT SETUP

We denote our method as student-oriented teacher training (*SoTeacher*), since it aims to learn the true label distribution to improve student performance, rather than to maximize teacher performance. We compare our method with the standard practice for teacher training in knowledge distillation (*Standard*) (*i.e.*, Eq. (1)). We conduct experiments on benchmark datasets including CIFAR-100 (Krizhevsky, 2009), Tiny-ImageNet (Tin, 2017), and ImageNet (Deng et al., 2009). We experiment with various backbone networks including ResNet (He et al., 2016), Wide ResNet (Zagoruyko & Komodakis, 2016) and ShuffleNet (Zhang et al., 2018b; Tan et al., 2019). We test the applicability of *SoTeacher* from different aspects of model compression in knowledge distillation including reduction of the width or depth, and distillation between heterogeneous neural architectures.

For knowledge distillation algorithms, we experiment with the original knowledge distillation method (KD) (Hinton et al., 2015), and a wide variety of other sophisticated knowledge distillation algorithms (see Sect. 5.2). We report the classification accuracies on the test set of both teacher and the student distilled from it. All results except ImageNet are presented with mean and standard deviation based on 3 independent runs. For Tiny-ImageNet, we also report the top-5 accuracy. For hyperparameters, we set $\lambda_{\mathrm{CR}}^{\max} = 1$ for both datasets, $\lambda_{\mathrm{LR}} = 10^{-5}$ for CIFAR-100 and $\lambda_{\mathrm{LR}} = 10^{-6}$

for Tiny-ImageNet/ImageNet. More detailed hyperparameter settings for neural network training and knowledge distillation can be found in Appendix D.

Table 1: Test accuracy of the teacher and student with knowledge distillation conducted on CIFAR-100. *SoTeacher* achieves better student accuracy than Standard for various architectures, depsite a lower teacher accuracy.

| | WRN40-2/WRN40-1 | | WRN40-2/WRN16-2 | | ResNet32x4/ShuffleNetV2 | |
|---|---|---|---|---|---|---|
| | **Student** | Teacher | **Student** | Teacher | **Student** | Teacher |
| Standard | $73.73 \pm 0.13$ | $76.38 \pm 0.13$ | $74.87 \pm 0.45$ | $76.38 \pm 0.13$ | $74.86 \pm 0.18$ | $79.22 \pm 0.03$ |
| *SoTeacher* | $\mathbf{74.35} \pm 0.23$ | $74.95 \pm 0.28$ | $\mathbf{75.39} \pm 0.23$ | $74.95 \pm 0.28$ | $\mathbf{77.24} \pm 0.09$ | $78.49 \pm 0.09$ |
| no-CR | $74.34 \pm 0.11$ | $74.30 \pm 0.12$ | $75.20 \pm 0.24$ | $74.30 \pm 0.12$ | $76.52 \pm 0.52$ | $77.73 \pm 0.17$ |
| no-LR | $73.81 \pm 0.15$ | $76.71 \pm 0.16$ | $75.21 \pm 0.13$ | $76.71 \pm 0.16$ | $76.23 \pm 0.18$ | $80.01 \pm 0.18$ |

Table 2: Test accuracy of the teacher and student with knowledge distillation conducted on Tiny-ImageNet and ImageNet. The student network is ResNet18, while the teacher network is ResNet34 for Tiny-ImageNet and ResNet152 for ImageNet. Due to computation constraints, we are not able to perform ablation experiments on ImageNet.

| | Tiny-ImageNet | | | | ImageNet | |
|---|---|---|---|---|---|---|
| | **Student** (Top-1) | **Student** (Top-5) | Teacher (Top-1) | Teacher (Top-5) | **Student** (Top-1) | Teacher (Top-1) |
| Standard | $66.19 \pm 0.17$ | $85.74 \pm 0.21$ | $64.94 \pm 0.32$ | $84.33 \pm 0.40$ | 71.30 | 77.87 |
| *SoTeacher* | $\mathbf{66.83} \pm 0.20$ | $86.19 \pm 0.22$ | $64.88 \pm 0.48$ | $84.91 \pm 0.41$ | 71.45 | 77.11 |
| no-CR | $66.39 \pm 0.27$ | $86.05 \pm 0.17$ | $64.36 \pm 0.43$ | $84.10 \pm 0.27$ | - | - |
| no-LR | $66.48 \pm 0.43$ | $\mathbf{86.20} \pm 0.40$ | $64.26 \pm 1.54$ | $84.48 \pm 0.84$ | - | - |

## 5.2 RESULTS

**End-to-end Knowledge Distillation Performance.**    Tables 1 and 2 show the evaluation results on CIFAR-100 and Tiny-ImageNet/ImageNet, respectively. Our teacher training method SoTeacher can improve the student's test accuracy consistently across different datasets and teacher/student architecture pairs. Note that the success of our teacher training method is not due to the high accuracy of the teacher. In Tables 1 and 2, one may already notice that our regularization method will hurt the accuracy of the teacher, despite that it can improve the accuracy of the student distilled from it.

On ImageNet particularly, the gain in student's accuracy by *SoTeacher* is minor, potentially because the large size of the dataset can already facilitate the learning of true label distribution as suggested by our theory. Nevertheless, one may observe that the teacher's accuracy is significantly lower when using *SoTeacher*, which implicitly suggests our teacher regularization method is tailored towards the student's accuracy.

Table 3: Estimation of the uncertainty quality of the teacher network trained by Standard and *SoTeacher*. The uncertainty quality is estimated by the ECE and NLL both before and after temperature scaling (TS).

| Dataset | Teacher | Method | ECE | NLL | ECE (w/ TS) | NLL (w/ TS) |
|---|---|---|---|---|---|---|
| CIFAR-100 | WRN40-2 | Standard | $0.113 \pm 0.003$ | $1.047 \pm 0.007$ | $0.028 \pm 0.004$ | $0.905 \pm 0.008$ |
| | | *SoTeacher* | $0.057 \pm 0.002$ | $0.911 \pm 0.013$ | $0.016 \pm 0.003$ | $0.876 \pm 0.012$ |
| CIFAR-100 | ResNet32x4 | Standard | $0.083 \pm 0.003$ | $0.871 \pm 0.010$ | $0.036 \pm 0.001$ | $0.815 \pm 0.008$ |
| | | *SoTeacher* | $0.037 \pm 0.001$ | $0.777 \pm 0.006$ | $0.021 \pm 0.001$ | $0.764 \pm 0.003$ |
| Tiny-ImageNet | ResNet34 | Standard | $0.107 \pm 0.007$ | $1.601 \pm 0.037$ | $0.043 \pm 0.002$ | $1.574 \pm 0.015$ |
| | | *SoTeacher* | $0.070 \pm 0.007$ | $1.496 \pm 0.031$ | $0.028 \pm 0.002$ | $1.505 \pm 0.033$ |

**Ablation Study.**    We toggle off the Lipschitz regularization (LR) or consistency regularization (CR) in *SoTeacher* (denoted as *no-LR* and *no-CR*, respectively) to explore their individual effects. As shown in Tables 1 and 2, LR and CR can both improve the performance individually. But on average, *SoTeacher* achieves the best performance when combining both LR and CR, as also demonstrated

in our theoretical analyses. Note that in Table 2, using Lipschitz regularization is not particularly effective because the regularization weight might not be properly tuned (see Figure 2).

**Quality of True Distribution Approximation.**     To further interpret the success of our teacher training method, we show that our regularization can indeed improve the approximation of the true label distribution thus benefiting the student generalization. Directly measuring the quality of the true distribution approximation is infeasible as the true distribution is unknown for realistic datasets. Follow previous works (Menon et al., 2021), we instead *estimate* the approximation quality by reporting the Expected Calibration Error (ECE) (Guo et al., 2017) and NLL loss of the teacher on a holdout set with one-hot labels. Since scaling the teacher predictions in knowledge distillation can improve the uncertainty quality (Menon et al., 2021), we also report ECE and NLL after temperature scaling, where the optimal temperature is located on an additional holdout set (Guo et al., 2017). As shown in Table 3, our teacher training method can consistently improve the approximation quality for different datasets and teacher architectures.

**Effect of Hyperparameters.**     We conduct additional experiments on Tiny-ImageNet as an example to study the effect of two regularization terms introduced by our teacher training method. For Lipschitz regularization, we modulate the regularization weight $\lambda_{\mathrm{LR}}$. For consistency regularization, we try different maximum regularization weights $\lambda_{\mathrm{CR}}^{\max}$ and different weight schedulers including linear, cosine, cyclic, and piecewise curves. Detailed descriptions of these schedulers can be found in Appendix D. As shown in Figure 2, both Lipschitz and consistency regularizations can benefit the teacher training in terms of the student generalization consistently for different hyperparameter settings. This demonstrates that our regularizations are not sensitive to hyperparameter selection. Note that the hyperparameter chosen to report the results in Tables 1 and 2 might not be optimal since we didn't perform extensive hyperparameter search in fear of overfitting small datasets. It is thus possible to further boost the performance by careful hyperparameter tuning.

In particular, Figure 2(a) shows that, as Lipschitz regularization becomes stronger, the teacher accuracy constantly decreases while the student accuracy increases and converges. This demonstrates that excessively strong Lipschitz regularization hurts the performance of neural network training, but it can help student generalization in the knowledge distillation context.

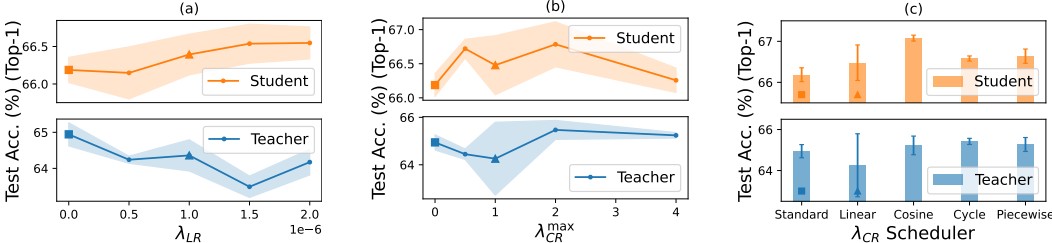

Figure 2: Effect of varying the hyperparameters in our teacher training method, including the weight for Lipschitz regularization $\lambda_{\mathrm{LR}}$, the weight for consistency regularization $\lambda_{\mathrm{CR}}$, and its scheduler. The settings of our method used to report the results (e.g. Table 2) are denoted as "▲". The standard teacher training practice is denoted as "■" for comparison.

**Other Knowledge Distillation Algorithms.**     Besides the original knowledge distillation algorithm, we experiment with various feature distillation algorithms including FitNets (Romero et al., 2015), AT (Zagoruyko & Komodakis, 2017), SP (Tung & Mori, 2019), CC (Peng et al., 2019), VID (Ahn et al., 2019), RKD (Park et al., 2019), PKT (Passalis & Tefas, 2018), AB (Heo et al., 2019), FT (Kim et al., 2018), NST (Huang & Wang, 2017), CRD (Tian et al., 2020) and SSKD (Xu et al., 2020). For the implementation of these algorithms, we refer to existing repositories for knowledge distillation (Tian et al., 2020; Shah et al., 2020; Matsubara, 2021) and author-provided codes. Although these distillation algorithms match all types of features instead of predictions between teacher and student, they will achieve the best distillation performance when combined with the prediction distillation (*i.e.* original KD). Therefore, our teacher training method should still benefit the effectiveness of these distillation algorithms. We also experiment with a curriculum distillation algorithm RCO (Jin et al., 2019) which distills from multiple checkpoints in teacher's training trajectory. Our teacher training method should also benefit RCO as the later teacher checkpoints become more student-oriented. As shown in Table 6,

our *SoTeacher* can boost the distillation performance of almost all these distillation algorithms, demonstrating its wide applicability.

**Student fidelity.**    Recent works have underlined the importance of student fidelity in knowledge distillation, namely the ability of the student to match teacher predictions (Stanton et al., 2021). Student fidelity can be viewed as a measure of knowledge distillation effectiveness that is orthogonal to student generalization, as the student is often unable to match the teacher predictions although its accuracy on unseen data improves (Furlanello et al., 2018; Mobahi et al., 2020). Here we measure the student fidelity by the average agreement between the student and teacher's top-1 predicted labels on the test set. As shown in Table 4, our teacher training method can consistently and significantly improve the student fidelity for different datasets and teacher-student pairs, which aligns with the improvement of student generalization shown by Table 1 and 2. This demonstrates that the teacher can better transfer its "knowledge" to the student with our training method.

Table 4: Average agreement (%) between the student and teacher's top-1 predictions on the test set.

| | CIFAR-100 | | | Tiny-ImageNet |
| --- | --- | --- | --- | --- |
| | WRN-40-2/WRN-40-1 | WRN-40-2/WRN-16-2 | ResNet32x4/ShuffleNetV2 | ResNet34/ResNet18 |
| Standard | $76.16 \pm 0.14$ | $76.92 \pm 0.29$ | $76.63 \pm 0.25$ | $71.33 \pm 0.07$ |
| *SoTeacher* | $\mathbf{77.92} \pm 0.27$ | $\mathbf{79.41} \pm 0.11$ | $\mathbf{80.36} \pm 0.13$ | $\mathbf{73.36} \pm 0.25$ |

## 6    RELATED WORK

**Understand knowledge distillation.**    There exist multiple perspectives in understanding the effectiveness of knowledge distillation. Besides the statistical perspective which views the soft prediction of the teacher as an approximation of the true label distribution (Menon et al., 2021; Dao et al., 2021), another line of work understands knowledge distillation from a regularization perspective, which views the teacher's soft predictions as instance-wise label smoothing (Yuan et al., 2020; Zhang & Sabuncu, 2020; Tang et al., 2020). More recently, (Allen-Zhu & Li, 2020) understands knowledge distillation from a feature learning perspective by focusing on the data that possesses a "multi-view" structure, that multiple features co-exist in the input and can be used to make the correct classification. Knowledge distillation is effective because the teacher can learn different features and transfer them to the student. Our theory is built on a similar assumption on the data structure, albeit we require the teacher to learn the true label distribution of the feature. Our theory thus can also be viewed as a bridge between the statistical perspective and feature learning perspective.

**Alleviate "teacher overfitting".**    Since in knowledge distillation, the distillation data is often reused from the teacher's training data, a teacher trained toward convergence is very likely to overfit its soft predictions on the distillation data. Intuitively, it is possible to tackle this problem by early stopping the teacher training (Cho & Hariharan, 2019). However, a meticulous hyperparameter search may be required since the epoch number to find the best checkpoint is often sensitive to the specific training setting such as the learning rate schedule. It is also possible to save multiple early teacher checkpoints for the student to be distilled from sequentially (Jin et al., 2019). Additionally, one can utilize a "cross-fitting" procedure to prevent the teacher from memorizing the training data. Namely, the training data is first partitioned into several folds, where the teacher predictions on each fold are generated by the teacher trained only on out-of-fold data (Dao et al., 2021). One can also train the teacher network jointly with student's network blocks, which imposes a regularization toward the student performance (Park et al., 2021). Different from these attempts, we train the teacher to directly learn the true label distribution of its training data, leading to a simple and practical student-oriented teacher training framework with minimum computation overhead.

## 7    CONCLUSION AND FUTURE WORK

In this work, we rigorously studied the feasibility to learn the true label distribution of the training data under a standard empirical risk minimization framework. We also explore possible improvements of current teacher training that facilitate such learning. In the future, we plan to adapt our theory to other knowledge distillation scenarios such as transfer learning and mixture of experts, and explore more effective student-oriented teacher network training methods.

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

# A  PROOF

## A.1  MODIFIED SOFTMAX

We provide more details for the modified Softmax we defined. Recall that

**Definition A.1** (Modified Softmax).

$$\widetilde{Softmax}(\hat{h}) = \frac{\exp(\frac{1}{M}\sum_m \hat{h}_m)}{(\prod_m \sum_k \exp(\hat{h}_{m,k}))^{1/M}}. \tag{8}$$

Our modified Softmax layer maps $\hat{h} \in \mathbb{R}^{M \times K}$ to the probabilistic output $f(x) \in \mathbb{R}^K$. It can be viewed as a combination of an average pooling layer and a Softmax layer, where the denominator of the softmax layer is a geometric average of the sum-exp of the probabilistic output of each batch $\hat{h}_m$.

## A.2  LEMMA 3.3

*Proof.* Given the labeled dataset $\mathcal{D} = \{(x^{(i)}, y^{(i)})\}$, the empirical risk of the network function $f$ under cross-entropy loss can be written as

$$
\begin{aligned}
&-\frac{1}{N}\sum_i \mathbb{1}(y^{(i)}) \cdot \log f(x^{(i)}) \\
=&-\frac{1}{N}\sum_i \mathbb{1}(y^{(i)}) \cdot \log \widetilde{Softmax}(w_C f_E(x^{(i)})) \\
:=&-\frac{1}{N}\sum_i \mathbb{1}(y^{(i)}) \cdot \log \widetilde{Softmax}\left(\hat{h}^{(i)}\right) \\
=&-\frac{1}{N}\sum_i \mathbb{1}(y^{(i)}) \cdot \log \frac{\exp(\frac{1}{M}\sum_m \hat{h}_m^{(i)})}{(\prod_m \sum_k \exp(\hat{h}_{m,k}^{(i)}))^{1/M}} \\
=&-\frac{1}{N}\sum_i \mathbb{1}(y^{(i)}) \cdot \log \left(\frac{\prod_m \exp \hat{h}_m^{(i)}}{\prod_m \sum_k \exp(\hat{h}_{m,k}^{(i)})}\right)^{1/M} \\
=&-\frac{1}{MN}\sum_i \mathbb{1}(y^{(i)}) \cdot \log \frac{\prod_m \exp(\hat{h}_m^{(i)})}{\prod_m \sum_k \exp(\hat{h}_{m,k}^{(i)})} \\
=&-\frac{1}{MN}\sum_i \mathbb{1}(y^{(i)}) \cdot \sum_m \log \frac{\exp(\hat{h}_m^{(i)})}{\sum_k \exp(\hat{h}_{m,k}^{(i)})} \\
=&-\frac{1}{MN}\sum_i \mathbb{1}(y^{(i)}) \cdot \sum_m \log Softmax(\hat{h}_m^{(i)}) \\
:=&-\frac{1}{MN}\sum_i \mathbb{1}(y^{(i)}) \cdot \sum_m \log p_m^{(i)},
\end{aligned} \tag{9}
$$

where we have defined $p_m^{(i)} = Softmax(\hat{h}_m^{(i)}) \equiv Softmax(w_c f_E(x_m^{(i)}))$. Here $x_m^{(i)}$ indicates the $m$-th patch of the input $x$.

Now recall our assumption that $f_E$ is an invariant feature extractor, which means that for any $i$ and $j$, any $m$ and $m'$, and any $\gamma$ and $\gamma'$, $f_E(\gamma(x_m^{(i)})) = f_E(\gamma'(x_m^{(j)}))$ as long as $z_m = z_{m'}$. Since, $p_m^{(i)} = Softmax(\hat{h}_m^{(i)}) = Softmax(w_C f_E(\gamma(x_m^{(i)})))$, this means that $p_m^{(i)} = p_{m'}^{(j)}$ as long as $z_m = z_{m'}$.

Therefore, the empirical risk can be regrouped as

$$-\frac{1}{MN}\sum_i 1(y^{(i)}) \cdot \sum_m \log p_m^{(i)}$$

$$= -\frac{1}{MN}\sum_i 1(y^{(i)}) \cdot \sum_{z\in\mathcal{Z}^i} \log p_z$$

$$= -\frac{1}{M}\sum_{z\in\mathcal{Z}} \left(\frac{1}{N}\sum_{\{i|z\in\mathcal{Z}^{(i)}\}} 1(y^{(i)})\right) \cdot \log p_z,$$

where we have let $p_z := p_m^{(i)}$ where $z_m = z$.

Now we use the KKT condition to approach the empirical risk minimization problem since the only variable is $p_z$. The first-order stationarity condition gives

$$\nabla_{p_z} \cdot = -\frac{1}{MN}\sum_i 1(y^{(i)}) \odot \frac{1}{p_z} + \lambda \mathbf{1} = 0, \tag{10}$$

where $\odot$ indicates the Hadamard product. Now Hadamard product both sides by $p_z$ we have

$$-\frac{1}{MN}\sum_i 1(y^{(i)}) + \lambda p_z = 0,$$

and thus

$$p_z = \frac{1}{MN\lambda}\sum_i 1(y^{(i)}).$$

Now consider the condition $\mathbf{1} \cdot p_z = 1$, we have

$$\frac{1}{MN\lambda}\sum_i 1(y^{(i)}) \cdot \mathbf{1} = 1.$$

Since $\sum_i 1(y^{(i)}) \cdot \mathbf{1} = |\{i|z \in \mathcal{Z}^{(i)}\}|$,

$$\lambda = \frac{|\{i|z \in \mathcal{Z}^{(i)}\}|}{MN}.$$

This leads to

$$p_z = \frac{1}{N_z}\sum_{\{i|z\in\mathcal{Z}^{(i)}\}} 1(y^{(i)}),$$

where $N_z := |\{i|z \in \mathcal{Z}^{(i)}\}|$.

$\square$

### A.3 LEMMA 3.4

*Proof.* The sum of independently but non-identically distributed multinomial random variables is known as the Possion multinomial distribution (PMD). We utilize a known result of PMD to prove this lemma.

**Proposition A.2** (Lin et al. (2022)). *Let $\boldsymbol{I}_i = (I_{i1,\cdots,I_{im}}), i = 1, \cdots, n$ be $n$ independent random indicators where $I_{ij} \in \{0,1\}$ and $\sum_{j=1}^m I_{ij} = 1$ for each $i$. Let $\boldsymbol{p}_i = (p_{i1}, \cdots, p_{im})$ be the probability vector that $\boldsymbol{I}_i$ is sampled from, where $\sum_{j=1}^m p_{ij} = 1$. Let $\boldsymbol{X}$ be the sum of these $n$ random indicators, namely $\boldsymbol{X} = (X_1, \cdots, X_m) = \sum_{i=1}^n \boldsymbol{I}_i$. Then we have*

$$\mathbb{E}[\boldsymbol{X}] = (p_{\cdot 1}, \cdots, p_{\cdot m}), \tag{11}$$

*where $p_{\cdot j} = \sum_{i=1}^n p_{ij}$.*

*And*

$$\Sigma_{jk} = \begin{cases} \sum_{i=1}^n p_{ij}(1 - p_{ij}), & \text{if } j = k, \\ -\sum_{i=1}^n p_{ij}p_{ik}, & \text{if } j \neq k, \end{cases} \tag{12}$$

*where $\boldsymbol{\Sigma}$ is the covariance matrix of $\boldsymbol{X}$.*

Following this result, since $1(y^{(i)})$ is a random indicator and $y^{(i)} \sim p_Y(\cdot|x^{(i)})$, we have

$$\mathbb{E}[\bar{y}_z] = \bar{p}(\cdot|z) := \frac{1}{N} \sum_{\{i|z \in \mathcal{Z}^{(i)}\}} p_Y(\cdot|x^{(i)}). \tag{13}$$

And let the covariance matrix of $\bar{y}_z$ to be $\Sigma_{\bar{y}_z}$. Let $N_z := |\{i|z \in \mathcal{Z}^{(i)}\}|$ be the number of inputs that contain feature $z$, we have

$$\begin{aligned}
\text{tr}[\Sigma_{\bar{y}_z}] &= \frac{1}{N^2} \sum_{\{i|z \in \mathcal{Z}^{(i)}\}} \sum_{k=1}^{K} p_Y(k|x^{(i)})(1 - p_Y(k|x^{(i)})) \\
&= \frac{1}{N^2} \sum_{\{i|z \in \mathcal{Z}^{(i)}\}} \sum_{k=1}^{K} \left( 1 - p_Y^2(k|x^{(i)}) \right) \\
&\leq \frac{(K-1)N_z}{N^2},
\end{aligned} \tag{14}$$

where we utilized the fact that $\sum_k p_Y^2(k|x^{(i)}) \geq \frac{1}{K}$ by the Cauchy-Schwarz inequality.

Then by a multivariate Chebyshev inequality (see, *e.g.*, Chen (2007)), we have with probability $1 - \delta$,

$$\|\bar{y}_z - \bar{p}(\cdot|z)\|_2^2 \leq \frac{\text{tr}(\Sigma_{\bar{y}_z})}{\delta} \leq \frac{(K-1)N_z}{N^2} \cdot \frac{1}{\delta}. \tag{15}$$

Now finally, we derive an estimation of $N_z$. Recall that each input contains $M$ feature names sampled from the feature vocabulary $|\mathcal{Z}|$ subject to the distribution $p_Z$. Therefore,

$$\mathbb{E}\left[\frac{N_z}{N}\right] = P(z \in \mathcal{Z}^{(i)}) = 1 - (1 - p_Z(z))^M \approx M p_Z(z). \tag{16}$$

For simplicity, we assume $p_Z$ is a uniform distribution, which means $p_Z(z) \approx 1/|\mathcal{Z}|$, for any $z$. Note that this is not a necessary assumption since $N_z$ can also be trivially bounded by $N$. Therefore we have

$$\|\bar{y}_z - \bar{p}(\cdot|z)\|_2^2 \leq \frac{(K-1)M}{N|\mathcal{Z}|} \cdot \frac{1}{\delta}. \tag{17}$$

Recall that $M$ is the number of features (patches) in each input, and $|\mathcal{Z}|$ is the total number of features in the vocabulary. This result thus indicates a larger feature vocabulary can improve the approximation while a more complicated input can hurt the approximation.

$\square$

## A.4 LEMMA 3.5

*Proof.* To prove this lemma we need an additional weak assumption on the true label distribution of features that are "similar".

**Assumption A.3.** *Let $I(z, z')$ define the pointwise mutual information (PMI) between features $z$ and $z'$. Let $\psi$ be a concave and monotonically **decreasing** function. We assume*

$$D_{KL}\left(p_Y(\cdot|z) \,\|\, p_Y(\cdot|z')\right) = \psi\left(I(z, z')\right). \tag{18}$$

Eq. (18) indicates that when sampling the features in an input, if two features are more likely to be sampled together, their true label distribution should be more similar.

Given Eq. (18) and recall that

$$p_Y(\cdot|x^{(i)}) = \left( \prod_m p_Y(\cdot|z_m) \right)^{1/M} = \left( \prod_{z \in \mathcal{Z}^{(i)}} p_Y(\cdot|z) \right)^{1/M},$$

it can be shown that the difference between the true label distribution of the input that contains feature $z$ and the true label distribution of feature $z$ can be bounded. In specific, for $i \in \{i' | z \in \mathcal{Z}^{(i')}\}$, we have

$$
\begin{aligned}
D_{\text{KL}}\left(p_Y(\cdot|z) \,\middle\|\, p_Y(\cdot|x^{(i)})\right) &= -\sum_k p_Y(k|z) \log\left(\frac{p_Y(k|x^{(i)})}{p_Y(k|z)}\right) \\
&= -\frac{1}{M} \sum_k p_Y(k|z) \log\left(\prod_{z' \in \mathcal{Z}^{(i)}} \frac{p_Y(k|z')}{p_Y(k|z)}\right) \\
&= -\frac{1}{M} \sum_k p_Y(k|z) \sum_{z' \in \mathcal{Z}^{(i)}} \log\left(\frac{p_Y(k|z')}{p_Y(k|z)}\right) \\
&= -\frac{1}{M} \sum_{z' \in \mathcal{Z}^{(i)}} \sum_k p_Y(k|z) \log\left(\frac{p_Y(k|z')}{p_Y(k|z)}\right) \\
&= \frac{1}{M} \sum_{z' \in \mathcal{Z}^{(i)}} D_{\text{KL}}\left(p_Y(\cdot|z) \,\|\, p_Y(\cdot|z')\right) \\
&= \frac{1}{M} \sum_{z' \in \mathcal{Z}^{(i)}} \psi\left(I(z,z')\right), \\
&\leq \psi(\bar{I}(i,z)).
\end{aligned}
\tag{19}
$$

where $\bar{I}(i,z) = \frac{1}{M} \sum_{z' \in \mathcal{Z}^{(i)}} I(z,z')$ is the average PMI between feature $z$ and other features $z'$ in an input that contains $z$. Here we utilize Jensen's inequality on concave functions.

Now we can show that the average true label distribution of inputs that contain feature $z$ approximates the true label distribution of feature $z$. Recall the definition of average true label distribution of inputs that contain feature $z$ is $\bar{p}(\cdot|z) := \frac{1}{N} \sum_{\{i | z \in \mathcal{Z}^{(i)}\}} p_Y(\cdot|x^{(i)})$. And we have

$$
\begin{aligned}
\|\bar{p}(\cdot|z) - p_Y(\cdot|z)\|_1 &= \left\|\left(\frac{1}{N} \sum_{\{i | z \in \mathcal{Z}^{(i)}\}} p_Y(\cdot|x^{(i)})\right) - p_Y(\cdot|z)\right\|_1 \\
&\leq \frac{1}{N} \left\|\sum_{\{i | z \in \mathcal{Z}^{(i)}\}} \left(p_Y(\cdot|x^{(i)}) - p_Y(\cdot|z)\right)\right\|_1 \\
&\leq \frac{1}{N} \sum_{\{i | z \in \mathcal{Z}^{(i)}\}} \left\|p_Y(\cdot|x^{(i)}) - p_Y(\cdot|z)\right\|_1 \\
&\leq \frac{2^{1/2}}{N} \sum_{\{i | z \in \mathcal{Z}^{(i)}\}} D_{\text{KL}}^{1/2}\left(p_Y(\cdot|z) \,\|\, p_Y(\cdot|x^{(i)})\right) \\
&\leq \frac{2^{1/2}}{N} \sum_{\{i | z \in \mathcal{Z}^{(i)}\}} \psi^{1/2}(\bar{I}(i,z)) \\
&\leq \frac{2^{1/2} N_z}{N} \psi^{1/2}\left(\bar{I}(z)\right) \\
&\approx \frac{2^{1/2} M}{|\mathcal{Z}|} \psi^{1/2}\left(\bar{I}(z)\right),
\end{aligned}
\tag{20}
$$

where $\bar{I}(z) = \frac{1}{N_z} \min_{\{i | z \in \mathcal{Z}^{(i)}\}} \bar{I}(i,z)$ is the average PMI over all inputs that contain feature $z$.

$\square$

## A.5 THEOREM 3.6

*Proof.* Recall the probabilistic output of the minimizer of the empirical risk is

$$f^*(x^{(i)}) = \left( \prod_m p_m^{(i)} \right)^{1/M} = \left( \prod_{z \in \mathcal{Z}^{(i)}} p_z \right)^{1/M}, \tag{21}$$

where $p_z$ defined in Section A.2 is the network's probabilistic output of each feature $z$.

Note that for all $k$, $f(x^{(i)})(k) \leq 1$ and $p_Y(k|x^{(i)}) \leq 1$. Therefore,

$$
\begin{aligned}
\left\| f^*(x^{(i)}) - p_Y(\cdot|x^{(i)}) \right\|_2 &\leq \exp\left( \left\| \log(f^*(x^{(i)})) - \log(p_Y(\cdot|x^{(i)})) \right\|_2 \right) \\
&= \exp\left( \frac{1}{M} \left\| \sum_{z \in \mathcal{Z}^{(i)}} (\log p_z - \log p_Y(\cdot|z)) \right\|_2 \right) \\
&\leq \exp\left( \frac{1}{M} \sum_{z \in \mathcal{Z}^{(i)}} \| \log p_z - \log p_Y(\cdot|z) \|_2 \right) \\
&\leq \exp\left( \frac{1}{M} \sum_{z \in \mathcal{Z}^{(i)}} \beta \log \| p_z - p_Y(\cdot|z) \|_2 \right) \\
&\leq \left( \max_{z \in \mathcal{Z}^{(i)}} \| p_z - p_Y(\cdot|z) \|_2 \right)^\beta \\
&= \left( \max_{z \in \mathcal{Z}^{(i)}} \| \bar{y}_z - p_Y(\cdot|z) \|_2 \right)^\beta \\
&= \left( \max_{z \in \mathcal{Z}^{(i)}} (\| \bar{y}_z - \bar{p}(\cdot|z) \|_2 + \| \bar{p}(\cdot|z) - p_Y(\cdot|z) \|_2) \right)^\beta \\
&\leq \left( \max_{z \in \mathcal{Z}^{(i)}} (\| \bar{y}_z - \bar{p}(\cdot|z) \|_2 + \| \bar{p}(\cdot|z) - p_Y(\cdot|z) \|_1) \right)^\beta \\
&= \left( \sqrt{\frac{(K-1)M}{N|\mathcal{Z}|} \frac{1}{\delta}} + \frac{2^{1/2} M}{|\mathcal{Z}|} \psi^{1/2}(\bar{I}_{\min}) \right)^\beta,
\end{aligned}
\tag{22}
$$

where $\beta := \min(\min_k f(x^{(i)})(k), \min_k p_Y(k|x^{(i)}))$ and $\bar{I}_{\min} = \min_{z \in \mathcal{Z}^{(i)}} \bar{I}(z)$.

$\square$

## A.6 THEOREM 3.9

*Proof.* We first present a Lemma similar to Lemma 3.3, which shows that the probabilistic predictions of features will still converge to the sample mean of the labels where the corresponding inputs contain this feature, up to some constant error.

**Lemma A.4** (Convergence of the probabilistic predictions of features with Lipschitz-continuous and transformation-robust feature extractor). *Let $\bar{y}_z := \frac{1}{N} \sum_{\{i|z \in \mathcal{Z}^{(i)}\}} 1(y^{(i)})$, where $\mathcal{Z}^{(i)}$ denote the set of feature names in the $i$-th input, and thus $\{i|z \in \mathcal{Z}^{(i)}\}$ denotes the set of inputs that contain feature $z$. Let $f^*$ be a minimizer of the empirical risk (Eq. (1)) and assume $f_E^*$ is a $L_X$-Lipschitz-continuous and $L_\Gamma$-transformation-robust feature extractor. Let $p_{f^*}(x_z) := Softmax(w_C f_E^*(x_z))$. We have with probability $1 - \delta$,*

$$\| p_{f^*}(x_z) - \bar{y}_z \| \leq L_z, \tag{23}$$

*where $L_z := 2L_\Gamma + L_X \tilde{O}(\frac{\nu_z}{\delta})$.*

*Proof.* Since $f_E^*$ is a $L_X$-Lipschitz-continuous and $L_\Gamma$-transformation-robust feature extractor, for any $i, j$, for any $m, m'$ and any $\gamma, \gamma'$, as long as $z_m = z_{m'} \equiv z$, we will have ,

$$
\begin{aligned}
&\|f_E(x_m^{(i)}) - f_E(x_{m'}^{(j)})\| \\
=&\|f_E(\gamma(x_z^{(i)})) - f_E(\gamma'(x_z^{(j)}))\| \\
\leq&\|f_E(\gamma(x_z^{(i)})) - f_E(x_z^{(i)})\| + \|f_E(\gamma'(x_z^{(j)})) - f_E(x_z^{(j)})\| + \|f_E(x_z^{(i)}) - f_E(x_z^{(j)})\| \\
\leq&2L_\Gamma + L_X\|x_z^{(i)} - x_z^{(j)}\|
\end{aligned}
\tag{24}
$$

Now since the feature representation in the input space is always concentrated, by Chebyshev's inequality we will have with probability $1 - \delta$

$$\|x_z^{(i)} - x_z^{(j)}\| \leq \tilde{O}\left(\frac{\nu_z}{\delta}\right).$$

Therefore, we have with probability $1 - \delta$,

$$\|f_E(x_m^{(i)}) - f_E(x_{m'}^{(j)})\| \leq L_z,$$

where $L_z := 2L_\Gamma + L_X\tilde{O}\left(\frac{\nu_z}{\delta}\right)$.

The empirical loss minimization now becomes

$$
\min_f -\frac{1}{MN} \sum_i 1(y^{(i)}) \cdot \sum_m \log p_m^{(i)}
\tag{25}
$$
$$
s.t. \quad \|p_m^{(i)} - p_{m'}^{(j)}\| \leq L_z, \text{ if } z_m = z_{m'}.
$$

Let $\mathcal{S}_z := \{i | z \in \mathcal{Z}^{(i)}\}$ for simplicity, we can have

$$
\begin{aligned}
-\frac{1}{MN} \sum_i 1(y^{(i)}) \cdot \sum_m \log p_m^{(i)} &= -\frac{1}{MN} \sum_i 1(y^{(i)}) \cdot \sum_{z \in \mathcal{Z}^{(i)}} \log p_z^{(i)} \\
&= -\frac{1}{MN} \sum_{z \in \mathcal{Z}} \sum_{\{i | z \in \mathcal{Z}^{(i)}\}} 1(y^{(i)}) \cdot \log p_z^{(i)}, \\
&= -\frac{1}{MN} \sum_{z \in \mathcal{Z}} \sum_{i \in \mathcal{S}_z} 1(y^{(i)}) \cdot \log p_z^{(i)},
\end{aligned}
\tag{26}
$$

Note that this basically means that we assign the label of an example $y^{(i)}$ to each feature $z$ contained in this example's input.

Therefore, (25) will be equivalent to the following problem.

$$
\min_{\{p_z^{(i)}\}} -\frac{1}{MN} \sum_{z \in \mathcal{Z}} \sum_{i \in \mathcal{S}_z} 1(y^{(i)}) \cdot \log p_z^{(i)}
$$
$$
s.t. \quad \|p_z^{(i)} - p_z^{(j)}\| \leq L_z, \mathbf{1} \cdot p_z^{(i)} = 1, \forall z, \forall i, j \in \mathcal{S}_z, i \neq j.
$$

Note that the constraint $\|\cdot\| \leq L_z$ is only imposed in each subset $\mathcal{S}_z$.

Using the KKT condition, the above problem can be further formulated as follows.

$$
\begin{aligned}
\min_{\{p_z^{(i)}\}} &-\frac{1}{MN} \sum_{z \in \mathcal{Z}} \sum_{i \in \mathcal{S}_z} 1(y^{(i)}) \cdot \log p_z^{(i)} \\
&+ \sum_z \sum_{j,k \in \mathcal{S}_z, j \neq k} \mu_{z;jk} \left(\|p_z^{(j)} - p_z^{(k)}\| - L_z\right) \\
&+ \sum_z \sum_{l \in \mathcal{S}_z} \lambda_{z;l}(\mathbf{1} \cdot p_z^{(l)} - 1), \\
s.t. \quad &\mu_{z;jk}(\|p_z^{(j)} - p_z^{(k)}\| - L_z) = 0, \ \mu_{z;jk} \geq 0, \ \forall z, \ \forall j, k \in \mathcal{S}_z, j \neq k, \\
&\|p_z^{(j)} - p_z^{(k)}\| \leq L_z, \ \mathbf{1} \cdot p_z^{(l)} = 1, \ \forall z, \ \forall j, k, l \in \mathcal{S}_z, j \neq k.
\end{aligned}
\tag{27}
$$

Using the first-order stationarity condition we have, $\forall z, \forall i \in \mathcal{S}_z$,

$$\nabla_{p_z^{(i)}} \cdot = -\frac{1}{MN} 1(y^{(i)}) \odot \frac{1}{p_z^{(i)}} + 2 \sum_{k \neq i} \mu_{ik}(p_z^{(i)} - p_z^{(k)}) + \lambda_i \mathbf{1} = 0, \tag{28}$$

where for simplicity we neglected the subscript $z$ since the condition is the same for all $z$.

Sum (28) over $i$ we have

$$-\frac{1}{MN} \sum_i 1(y^{(i)}) \odot \frac{1}{p_z^{(i)}} + 2 \sum_i \sum_{k \neq i} \mu_{ik}(p_z^{(i)} - p_z^{(k)}) + \sum_i \lambda_i \mathbf{1} = 0. \tag{29}$$

Note that $\sum_i \sum_{k \neq i} \mu_{ik}(p_z^{(i)} - p_z^{(k)}) = 0$ since each pair of $i, k$ appears twice in the sum, where $\mu_{ik}$ is the same but the sign of $p_z^{(i)} - p_z^{(k)}$ is different. Therefore

$$-\frac{1}{MN} \sum_i 1(y^{(i)}) \odot \frac{1}{p_z^{(i)}} + \sum_i \lambda_i \mathbf{1} = 0. \tag{30}$$

Notice that to minimize the loss, if $y^{(i)} = y^{(j)}$, it is necessary that $p_z^{(i)} = p_z^{(j)}$.

Now we dot product both sides of (30) with $1(k)$, we have

$$-\frac{1}{MN} \frac{|\{i|y^{(i)} = k\}|}{p_z^{(i)}[k]} + \sum_i \lambda_i = 0, \ \forall i \in \{i|y^{(i)} = k\},$$

which means that

$$p_z^{(i)}[k] = \frac{|\{i|y^{(i)} = k\}|}{MN \sum_i \lambda_i}, \text{ if } y^{(i)} = k.$$

Now recall the constraint that when $i \neq j$,

$$\|p_z^{(i)} - p_z^{(j)}\| \leq L_z.$$

This at least indicates that

$$\|p_z^{(i)} - p_z^{(j)}\|_\infty \leq L_z.$$

Therefore,

$$p_z^{(i)}[k'] - \frac{|\{i|y^{(i)} = k'\}|}{MN \sum_i \lambda_i} \leq L_z, \text{ if } k' \neq y^{(i)}.$$

This implies that

$$\left\| p_z^{(i)} - \frac{1}{MN \sum_i \lambda_i} \sum_i 1(y^{(i)}) \right\| \leq L_z$$

Now consider the condition $\mathbf{1} \cdot p_z^{(i)} = 1$, we will have

$$\sum_i \lambda_i = \frac{N_z}{MN}.$$

Thus

$$\left\| p_z^{(i)} - \bar{y}_z \right\| \leq L_z,$$

where we recall $\bar{y}_z = \frac{1}{N_z} \sum_i 1(y^{(i)})$. $\qquad\square$

Now to prove Theorem 3.9, we only need to combine Lemma A.4 and Lemmas 3.4, 3.5, where the reasoning is exactly same as the proof of Theorem 3.6.

$\qquad\square$

# B   LIMITATIONS

In this paper we focus on the theoretical feasibility of learning the true label distribution of training examples with empirical risk minimization. Therefore we only analyze the existence of such a desired minimizer, but neglect the optimization process to achieve it. By explore the optimization towards true label distribution, potentially more dynamics can be found to inspire new regularization techniques.

Also, our proposed method for training a student-oriented teacher may not be able to advance the state-of-the-art significantly, as the regularization techniques based inspired by our analyses (e.g., Lipschitz regularization and consistency regularization) may more or less be leveraged by existing training practice of deep neural networks, either implicitly or explicitly.

# C   IMPLEMENTATION DETAILS

**Lipschitz regularization.**    Following previous practice using Lipschitz regularization for generalization on unseen data (Yoshida & Miyato, 2017) or stabilizing generative model (Miyato et al., 2018), we regularize the Lipschitz constant of a network by constraining the Lipschitz constant of each trainable component. The regularization term is thus defined as $\ell_{\text{LR}} = \sum_f \text{Lip}(f)$, where $f$ denotes a trainable component in the network $\boldsymbol{f}$. The Lipschitz constant of a network component $\text{Lip}(f)$ induced by a norm $\|\cdot\|$ is the smallest value $L$ such that for any input features $h, h'$, $\|f(h) - f(h')\| \leq L\|h - h'\|$. Here we adopt the Lipschitz constant induced by 1-norm, since its calculation is accurate, simple and efficient. For calculating the Lipschitz constants of common trainable components in deep neural networks, we refer to (Gouk et al., 2021) for a comprehensive study.

**Consistency regularization.**        We design our consistency regularization term as $\ell_{\text{CR}} = \frac{1}{N} \sum_i \left\| \boldsymbol{f}(x_i) - \overline{\boldsymbol{f}(x_i)} \right\|_2^2$, where we follow previous work (Laine & Aila, 2017) and employ MSE to penalize the difference. Here $\overline{\boldsymbol{f}(x)}$ is the aggregated prediction of an input $x$, which we calculate as the simple average of previous predictions $\overline{\boldsymbol{f}(x)}_t = \frac{1}{t} \sum_{t'=0}^{t-1} \boldsymbol{f}(x)_{t'}$, where we omit the data augmentation operator for simplicity. At epoch 0 we simply skip the consistency regularization. Note that such a prediction average can be implemented in an online manner thus there is no need to store every previous prediction of an input.

# D   DETAILS OF EXPERIMENT SETTING

## D.1   HYPERPARAMETER SETTING FOR TEACHER NETWORK TRAINING

For all the experiments on CIFAR-100, we employ SGD as the optimizer and train for 240 epochs with a batch size of 64. The learning rate is initialized at 0.05 and decayed by a factor of 10 at the epochs 150, 180 and 210, with an exception for ShuffleNet where the learning rate is initialized at 0.01 following existing practice (Tian et al., 2020; Park et al., 2021). The weight decay and momentum are fixed as 0.0005 and 0.9 respectively. The training images are augmented with random cropping and random horizontal flipping with a probability of 0.5.

For Tiny-ImageNet experiments, we employ SGD as the optimizer and conduct the teacher training for 90 epochs with a batch size of 128. The learning rate starts at 0.1 and is decayed by a factor of 10 at epochs 30 and 60. The weight decay and momentum are fixed as 0.0005 and 0.9 respectively. The training images are augmented with random rotation with a maximum degree of 20, random cropping and random horizontal flipping with a probability of 0.5. For student training the only difference is that we train for additional 10 epochs, with one more learning rate decay at epoch 90, aligned with previous settings (Tian et al., 2020).

For consistency regularization in our teacher training method, we experiment with various weight schedules besides the linear schedule mentioned in the main paper. We list the formulas for these schedules in the following. Here $t$ denotes the epoch number, $T$ denotes the total number of epochs, and $\lambda_{CR}^{\max}$ denotes the maximum weight.

Table 5: $\beta$ for different feature distillation algorithms

|         | $\beta$ | | | $\beta$ | | | $\beta$ | |
|         | Standard | SoTeacher | | Standard | SoTeacher | | Standard | SoTeacher |
|---------|----------|-----------|-----|----------|-----------|-----|----------|-----------|
| FitNets | 100      | 50        | AT  | 1000     | 500       | SP  | 3000     | 1500      |
| CC      | 0.02     | 0.01      | VID | 1.0      | 0.5       | RKD | 1.0      | 0.5       |
| PKT     | 30000    | 15000     | AB  | 1.0      | 0.5       | FT  | 200      | 100       |
| NST     | 50       | 25        | CRD | 0.8      | 0.5       |     |          |           |

- Cosine schedule:

$$\lambda_{CR}(t) = \cos\left[\left(1 - \frac{t}{T}\right)\frac{\pi}{2}\right]\lambda_{CR}^{\max}$$

- Cyclic schedule:

$$\lambda_{CR}(t) = \sqrt{1 - \left(1 - \frac{t}{T}\right)^2}\,\lambda_{CR}^{\max}$$

- Piecewise schedule:

$$\lambda_{CR}(t) = \begin{cases} 0, & 0 < t \le T/3, \\ \lambda_{CR}^{\max}/2, & T/3 < t \le 2T/3, \\ \lambda_{CR}^{\max}, & 2T/3 < t \le T. \end{cases}$$

## D.2 HYPERPARAMETER SETTING FOR KNOWLEDGE DISTILLATION ALGORITHMS

For knowledge distillation algorithms we refer to the setting in RepDistiller [2]. Specifically, for original KD, the loss function used for student training is defined as

$$\ell = \alpha\ell_{\text{Cross-Entropy}} + (1 - \alpha)\ell_{KD}.$$

We grid search the best hyper-parameters that achieve the optimal performance, namely the loss scaling ratio $\alpha$ is set as $0.5$ and the temperature is set as $4$ for both CIFAR-100 and Tiny-ImageNet. For all feature distillation methods combined with KD the loss function can be summarized as (Tian et al., 2020)

$$\ell = \gamma\ell_{\text{Cross-Entropy}} + \alpha\ell_{KD} + \beta\ell_{\text{Distill}},$$

where we grid search the optimal $\gamma$ and $\alpha$ to be $1.0$ and $1.0$ respectively. When using our teacher training method, all these hyperparameters are kept same except that for all feature distillation algorithms the scaling weights corresponding to the feature distillation losses $\beta$ are cut by half, as we wish to rely more on the original KD that is well supported by our theoretical understanding. Table 5 list $\beta$ used in our experiments for all feature distillation algorithms. For SSKD (Xu et al., 2020) the hyperparameters are set as $\lambda_1 = 1.0$, $\lambda_2 = 1.0$, $\lambda_3 = 2.7$, $\lambda_4 = 10.0$ for standard training and $\lambda_1 = 1.0$, $\lambda_2 = 1.0$, $\lambda_3 = 1.0$, $\lambda_4 = 10.0$ for our methods. For the curriculum distillation algorithm RCO we experiment based on one-stage EEI (equal epoch interval). We select 24 anchor points (or equivalently every 10 epochs) from the teacher's saved checkpoints.

## E ADDITIONAL EXPERIMENT RESULTS

**Training overhead.** Compared to standard teacher training, the computation overhead of *SoTeacher* is mainly due to the calculation of the Lipschitz constant, which is efficient as it only requires simple arithmetic calculations of the trainable weights of a neural network (see Section 4). Empirically we observe that training with *SoTeacher* is only slightly longer than the standard training for about $5\%$. The memory overhead of *SoTeacher* is incurred by buffering an average prediction for each input. However, since such prediction requires no gradient calculation we can simply store it in a memory-mapped file.

---

[2]https://github.com/HobbitLong/RepDistiller

Table 6: *SoTeacher* consistently outperforms Standard on CIFAR-100 with various KD algorithms.

| | WRN40-2/WRN40-1 | | WRN40-2/WRN16-2 | | ResNet32x4/ShuffleV2 | |
|---|---|---|---|---|---|---|
| | Standard | *SoTeacher* | Standard | *SoTeacher* | Standard | *SoTeacher* |
| FitNet | $74.06 \pm 0.20$ | $\mathbf{74.88} \pm 0.15$ | $75.42 \pm 0.38$ | $\mathbf{75.64} \pm 0.20$ | $76.56 \pm 0.15$ | $\mathbf{77.91} \pm 0.21$ |
| AT | $73.78 \pm 0.40$ | $\mathbf{75.12} \pm 0.17$ | $75.45 \pm 0.28$ | $\mathbf{75.88} \pm 0.09$ | $76.20 \pm 0.16$ | $\mathbf{77.93} \pm 0.15$ |
| SP | $73.54 \pm 0.20$ | $\mathbf{74.71} \pm 0.19$ | $74.67 \pm 0.37$ | $\mathbf{75.94} \pm 0.20$ | $75.94 \pm 0.16$ | $\mathbf{78.06} \pm 0.34$ |
| CC | $73.46 \pm 0.12$ | $\mathbf{74.76} \pm 0.16$ | $75.08 \pm 0.07$ | $\mathbf{75.67} \pm 0.39$ | $75.43 \pm 0.19$ | $\mathbf{77.68} \pm 0.28$ |
| VID | $73.88 \pm 0.30$ | $\mathbf{74.89} \pm 0.19$ | $75.11 \pm 0.07$ | $\mathbf{75.71} \pm 0.19$ | $75.95 \pm 0.11$ | $\mathbf{77.57} \pm 0.16$ |
| RKD | $73.41 \pm 0.47$ | $\mathbf{74.66} \pm 0.08$ | $75.16 \pm 0.21$ | $\mathbf{75.59} \pm 0.18$ | $75.28 \pm 0.11$ | $\mathbf{77.46} \pm 0.10$ |
| PKT | $74.14 \pm 0.43$ | $\mathbf{74.89} \pm 0.16$ | $75.45 \pm 0.09$ | $\mathbf{75.53} \pm 0.09$ | $75.72 \pm 0.18$ | $\mathbf{77.84} \pm 0.03$ |
| AB | $73.93 \pm 0.35$ | $\mathbf{74.86} \pm 0.10$ | $70.09 \pm 0.66$ | $\mathbf{70.38} \pm 0.87$ | $76.27 \pm 0.26$ | $\mathbf{78.05} \pm 0.21$ |
| FT | $73.80 \pm 0.15$ | $\mathbf{74.75} \pm 0.13$ | $75.19 \pm 0.15$ | $\mathbf{75.68} \pm 0.28$ | $76.42 \pm 0.17$ | $\mathbf{77.56} \pm 0.15$ |
| NST | $73.95 \pm 0.41$ | $\mathbf{74.74} \pm 0.14$ | $74.95 \pm 0.23$ | $\mathbf{75.68} \pm 0.16$ | $76.07 \pm 0.08$ | $\mathbf{77.71} \pm 0.10$ |
| CRD | $74.44 \pm 0.11$ | $\mathbf{75.06} \pm 0.37$ | $75.52 \pm 0.12$ | $\mathbf{75.95} \pm 0.02$ | $76.28 \pm 0.13$ | $\mathbf{78.09} \pm 0.13$ |
| SSKD | $75.82 \pm 0.22$ | $\mathbf{75.94} \pm 0.18$ | $76.31 \pm 0.07$ | $\mathbf{76.32} \pm 0.09$ | $78.49 \pm 0.10$ | $\mathbf{79.37} \pm 0.11$ |
| RCO | $74.50 \pm 0.32$ | $\mathbf{74.81} \pm 0.04$ | $75.24 \pm 0.34$ | $\mathbf{75.50} \pm 0.12$ | $76.75 \pm 0.13$ | $\mathbf{77.59} \pm 0.31$ |

Table 7: Performance of the knowledge distillation when training the teacher using existing regularization methods for learning quality uncertainty on unseen data.

| | WRN40-2/WRN40-1 | |
|---|---|---|
| | **Student** | Teacher |
| Standard | $73.73 \pm 0.13$ | $76.38 \pm 0.13$ |
| *SoTeacher* | $\mathbf{74.35} \pm 0.23$ | $74.95 \pm 0.28$ |
| $\ell_2 \ (5 \times 10^{-4})$ | $73.73 \pm 0.13$ | $76.38 \pm 0.13$ |
| $\ell_1 \ (10^{-5})$ | $73.60 \pm 0.15$ | $73.52 \pm 0.05$ |
| Mixup ($\alpha = 0.2$) | $73.19 \pm 0.21$ | $77.30 \pm 0.20$ |
| Cutmix ($\alpha = 0.2$) | $73.61 \pm 0.26$ | $78.42 \pm 0.07$ |
| Augmix ($\alpha = 1, k = 3$) | $73.83 \pm 0.09$ | $77.80 \pm 0.30$ |
| CRL ($\lambda = 1$) | $74.13 \pm 0.29$ | $76.69 \pm 0.16$ |

## F EXPERIMENTS WITH UNCERTAINTY REGULARIZATION METHODS ON UNSEEN DATA

**Uncertainty learning on unseen data.** Since the objective of our student-oriented teacher training is to learn label distributions of the training data, it is related to those methods aiming to learn quality uncertainty on the unseen data. We consider those methods that are feasible for large teacher network training, including (1) classic approaches to overcome overfitting such as $\ell_1$ and $\ell_2$ regularization, (2) modern regularizations such as label smoothing (Szegedy et al., 2016) and data augmentations such as mixup (Zhang et al., 2018a) and Augmix (Hendrycks et al., 2020), (3) Post-training methods such as temperature scaling (Guo et al., 2017), as well as (4) methods that incorporate uncertainty as a learning ojective such as confidence-aware learning (CRL) (Moon et al., 2020).

We have conducted experiments on CIFAR-100 using all these methods and the results can be found in Appendix F. Unfortunately, the performance of these regularization methods is unsatisfactory in knowledge distillation — only CRL can slightly outperform the standard training. We believe the reasons might be two-folds. First, most existing criteria for uncertainty quality on the unseen data such as calibration error (Naeini et al., 2015) or ranking error (Geifman et al., 2019), only require the model to output an uncertainty estimate that is correlated with the probability of prediction errors. Such criteria may not be translated into the approximation error to the true label distribution. Second, even if a model learns true label distribution on unseen data, it does not necessarily have to learn true label distribution on the training data, as deep neural networks tend to memorize the training data.

**Experiment setup.** We conduct experiments on CIFAR-100 with teacher-student pair WRN40-2/WRN40-1. We employ the original KD as the distillation algorithm. The hyperparameter settings are the same as those mentioned in the main results (see Appendix D). For each regularization method we grid search the hyperparameter that yields the best student performance. The results are summarized in Table 7.

**Classic regularization.**      We observe that with stronger $\ell_2$ or $\ell_1$ regularization the student performance will not deteriorate significantly as teacher converges. However, it also greatly reduces the performance of the teacher. Subsequently the performance of the student is not improved as shown in Table 7.

**Label smoothing.**     Label smoothing is shown to not only improve the performance but also the uncertainty estimates of deep neural networks (Müller et al., 2019). However, existing works have already shown that label smoothing can hurt the effectiveness of knowledge distillation (Müller et al., 2019), thus we neglect the results here. An intuitive explanation is that label smoothing encourages the representations of samples to lie in equally separated clusters, thus "erasing" the information encoding possible secondary classes in a sample (Müller et al., 2019).

**Data augmentation.**      Previous works have demonstrated that mixup-like data augmentation techniques can greatly improve the uncertainty estimation on unseen data (Thulasidasan et al., 2019; Hendrycks et al., 2020). For example, Mixup augmented the training samples as $x := \alpha x + (1 - \alpha)x'$, and $y := \alpha y + (1 - \alpha)y'$, where $(x', y')$ is a randomly drawn pair not necessarily belonging to the same class as $x$.

As shown in Table 7, stronger mixup can improve the performance of the teacher, whereas it can barely improve or even hurt the performance of the student. Based on our theoretical understanding of knowledge distillation, we conjecture the reason might be that mixup distorts the true label distribution of an input stochastically throughout the training, thus hampering the learning of true label distribution.

**Temperature scaling.**     Previous works have suggested using the uncertainty on a validation set to tune the temperature for knowledge distillation either in standard learning (Menon et al., 2021) or robust learning (Dong et al., 2021). However, the optimal temperature may not be well aligned with that selected based on uncertainty (Menon et al., 2021). We neglect the experiment results here as the distillation temperature in our experiments is already fine-tuned.

**Uncertainty learning.**      CRL designs the loss function as $\ell = \ell_{\text{CE}} + \lambda\ell_{\text{CRL}}$, where $\ell_{\text{CE}}$ is the cross-entropy loss and $\ell_{\text{CRL}}$ is an additional regularization term bearing the form of

$$\ell_{\text{CRL}} = \max\left(0, -g(c(x_i), c(x_j))(p(x_i) - p(x_j)) + |c(x_i) - c(x_j)|\right), \tag{31}$$

where $p(x) = \max_k \mathbf{f}(x)^k$ is the maximum probability of model's prediction on a training sample $x$ and

$$c(x) = \frac{1}{t - 1} \sum_{t'=1}^{t-1} 1(\arg\max_k \mathbf{f}(x)_{t'}^k = y)$$

is the frequency of correct predictions through the training up to the current epoch. Here $g(c_i, c_j) = 1$ if $c_i > c_j$ and $g(c_i, c_j) = -1$ otherwise. Although originally proposed to improve the uncertainty quality of deep neural networks in terms of ranking, we found that CRL with a proper hyperparameter can improve the distillation performance, as shown in Table 7.

We note that the effectiveness of CRL on distillation can be interpreted by our theoretical understanding, as its regularization term (31) is essentially a special form of consistency regularization. To see this we first notice (31) is a margin loss penalizing the difference between $p(x)$ and $c(x)$ in terms of ranking. We then rewrite $c(x)$ as

$$c(x) = \mathbf{1}_y \cdot \frac{1}{t - 1} \sum_{t'=1}^{t-1} \text{Onehot}[\mathbf{f}(x)_{t'}], \tag{32}$$

which is similar to our consistency regularization target, except the prediction is first converted into one-hot encoding. Such variation may not be the best for knowledge distillation as we wish those secondary class probabilities in the prediction be aligned as well.

