# OpenReview forum: "Toward Student-oriented Teacher Network Training for Knowledge Distillation"
_ICLR.cc/2024/Conference — ICLR 2024 poster_

### Official Review · Reviewer_Xgmp · 2023-10-28

**Soundness:** 2 fair
**Presentation:** 2 fair
**Contribution:** 2 fair
**Rating:** 5
**Confidence:** 3

**Summary:**

The authors investigates how to properly train a teacher model aimed at improving the student generalization performance, rather than the teacher generalization performance. Rooted in a theoretical analysis showing that a teacher model can indeed learn the underlying true label distribution they empirically propose to do so, by modifying the training objective of the teacher to include Lipschitz and consistency regularization.

**Strengths:**

- Research into how to train teachers explicitly aimed at improving student performance is important, and in large part orthogonal to most common distillation on improving knowledge distillation techniques.
- The authors provide both theoretical analyses and supporting empirical experiments.
- There are a sufficient amount of (diverse) experiments investigating the proposed approach.

**Weaknesses:**

I will separate my concerns into two parts; one on the theoretical analysis and one on the empirical experiments.

**Theoretical**
- The use of $\mathbb{E}$ for the *empirical* expectation is very unconventional, and I would strongly suggest to change this notation. This also conflicts with mentions of distributions later on. It is unclear when we are considering random variables and emprical observations.
- *"[...] of the distillation data, which is often reused from the teacher’s training data."* It appears as reusing the teacher's training data is an underlying assumption of the paper, which is a notable assumption, and should be stated explicitly as an assumption.
- *"In particular, minimization of Eq. (1) would lead to $f(x) = 1(y)$ for any input x, [...]"*. Why does this hold? There are no assumptions on $\ell(f(x), y)$ in (1), and the statement would not hold in general?
- Page 4 above (3). Is this the same softmax as other places? If not, why? If so, fix notation.
- *"[...] a continuous distribution with finite variance, namely $\mathrm{Var}[X|z] \leq \nu_z$"*. Note, $\mathrm{Var}[X|z] \leq \nu_z$ is *bounded* variance not only finite variance. What is the true assumption here?

**Empirical**
- The authors never compares to the conventional and very simple setting of early-stopping the teacher and/or carefully tuning the temperature scaling of the teacher predictions as is commonly done in practice.
- You both state that *"This demonstrates that our regularizations are not sensitive to hyperparameter selection."* and argue that *"It is thus
possible to further boost the performance by careful hyperparameter tuning"*, which seems contradicting.
- About Figure 2(a): *"[...] the teacher accuracy constantly decreases while the student accuracy increases and converges."* The teacher accuracy does not constantly decrease, but at best appear to have a downward trend. Furthermore, it is a stretch to say that the student accuracy converges due to no increase in 1 of 4 intervals. *"This demonstrates that excessively strong Lipschitz regularization hurts the performance of neural network training [...]"*. This is not supported by the figure where the teacher test accuracy for $\lambda_{LR} = 0.5 \cdot 10^{-6}$ and $\lambda_{LR} = 2 \cdot 10^{-6}$ is equal. Hence, the stronger regularization does not really affect the teacher performance.
- Throughout the paper, it is argued that a reduced teacher performance can yield improvements in the student generalization. However, from Figure 2(c)  at the performance of both the teacher and the student increase when using the cosine, cycle or piecewise schedules compared to the standard procedure. In fact, only the case of linear schedule (used elsewhere in the paper) show an increase in student performance combined with a decrease in teacher performance. Thus, it is unclear what the effect the experiments actual show? Is the improvement merely an artifact of other parts of the training scheme?
- There are no comparisons to Park et al. (2021), "Learning Student-Friendly Teacher Networks for Knowledge Distillation", although they provide experimental analysis on identical settings as this paper, and appear to notably outperform the proposed SoTeacher procedure. Granted, Park et al. (2021) propose a more involved procedure, but a comparison is needed.
- Section on Student fidelity: Care must be taken, when interpreting teacher-student agreement (as argued by Stanton (2021)). Improved agreement does not imply that the student learns better from the teacher. This can merely be an artifact of improved student performance (or decreased teacher performance).

**Minor**
- The introduction lacks reference to Mirzadeh et al (2020), "Improved Knowledge Distillation via Teacher Assistant" on the teacher-student gap.
- *"[...] namely $x_z := \gamma(x_z)$."* Overloaded notation here. Also, there is an inconsitent use of $x_{z_m}$ vs. $x_m$. For instance, it is unclear what $x_m$ in $h_m := f_E(x_m)$ refers to?
- *"[...] for any two inputs $i$ and $j$ [...]"* and other places. It is odd to refer to inputs by their indices. Replace by $x^{(i)}$ or write *"[...] for any two inputs indexed by $i$ and $j$ [...]"*
- Mention somewhere that the proofs are in the appendix.
- Table 1: Reconsider the "WRN40-2/WRN40-1" notation in the table, as it is not clear what is the teacher and student.
- The related works on "Understand knowledge distillation" miss references to strong theoretical works from Phuong and Lampert (2019), "Towards Understanding Knowledge Distillation", Borup and Andersen (2021), "Even your Teacher Needs Guidance: Ground-Truth Targets Dampen Regularization Imposed by Self-Distillation", and Mobahi et al. (2020), "Self-Distillation Amplifies Regularization in Hilbert Space".
- The related works on "Alleviate “teacher overfitting”." lacks reference to Dong (2019), "Distillation ≈ Early Stopping? Harvesting Dark Knowledge Utilizing Anisotropic Information Retrieval For Overparameterized Neural Network".

**Questions:**

- Figure 1: Why does a student distilled from a teacher checkpoint with 55-60% accuracy yield a test accuracy of about 71%, thereby exceeding the teacher significantly?
- Figure 1: It is known that early-stopping the teacher is beneficial, and SoTeacher does not improve over an early-stopped teacher (at 150 epochs). It is unclear why this approach is better than early stopping?

---

> ### Author Response · Authors · 2023-11-23
>
> Thank you for the valuable comments and suggestions. Please see our detailed response below.
>
> **___**
>
> ### ***Theoretical***
>
> __Q. Notation $\mathbb{E}$__
>
> Thanks for your suggestion. We will change the notation in the revision.
>
>
>
> __Q. Assumption on reusing the training data__
>
> We thank the reviewer for pointing this out. We will explicitly state this assumption in the revision.
>
>
>
> __Q. Assumption on the loss function__
>
> We agree with the reviewer that we may need mild assumptions on the loss function to let $f(x) = 1(y)$, although most widely used loss functions would satisfy this requirement. We thank the reviewer for pointing this out and will update accordingly in the revision.
>
>
>
> __Q. Notation softmax__
>
> Here the softmax should be our modified softmax. Thanks for pointing this out and we will revise accordingly.
>
>
>
> __Q. Assumption on the variance__
>
> We note that we will need bounded variance here. We thank the reviewer for pointing this out and we will revise accordingly.
>
>
> **___**
>
> ### ***Empirical***
>
> __Q. Compare with early stopping and other regularization methods (Also question 2)__
>
> We note that properly early stopping the teacher training may indeed be able to achieve student performance comparable to SoTeacher. However, as also mentioned in Line 356 in our paper, it is often difficult to locate the right epoch number for early stopping. Slight deviation from the best epoch number may significantly impair the student's performance. A meticulous search for the best epoch number will induce high computation costs as one may need to conduct full student training for each epoch number. In contrast, SoTeacher can achieve the best student performance even when the teacher fully converges, which will be much more efficient and friendly to use in practice.
>
> For other popular training or post-training techniques including classic regularizations such as l1 and l2, label smoothing, advanced data augmentation, temperature scaling, and uncertainty learning methods, we have provided a detailed comparison in Appendix F.
>
>
>
> __Q. Clarification of the statement__
>
> Here *"This demonstrates that our regularizations are not sensitive to hyperparameter selection."* refers to the analyses of hyperparameter sensitivity as shown in Figure 2, while *"It is thus possible to further boost the performance by careful hyperparameter tuning"* refers to the main result reported in Tables 1 and 2, where we didn't report the best result yielded by the hyperparameter search but instead report the result by a more intuitive hyperparameter selection.
>
>
>
> __Q. Clarification of Figure 2(a)__
>
> We agree with the reviewer that this figure may need more careful interpretation. Here we generally look at the trend of the curves as the hyperparameter value varies. We will add more grid search points in the revision.
>
>
>
> __Q. Clarification of the argument__
>
> We would like to clarify that we didn't argue reducing teacher performance can improve student performance in our paper. Instead, our argument is that strong teacher performance does not necessarily improve student performance. Therefore, we may need to tailor the teacher training such that the teacher's soft predictions better approximate the true label distribution. Such a goal does not necessarily imply an inferior teacher.
>
>
>
> __Q. Comparison to an existing method__
>
> We note that our proposed method is orthogonal to the method proposed by [1], as their method focuses on how to initialize the student networks, while we focus on how to train the teacher model for better student performance.
>
>
>
> __Q. Interpretation of the student fidelity__
>
> We agree with the reviewer that improved student fidelity does not necessarily imply that the student can better learn from the teacher. We incorporate such analyses as side evidence of the effectiveness of our method. We will modify our statement carefully in the revision.
>
>
>
> __Q. Phenomenon that student is better than teacher (question 1)__
>
> We note that it is possible that the student performance can exceed the teacher performance significantly in Figure 1. This is because in the distillation experiments here, the true labels are always used for training the student on top of the soft labels given by the teacher, and the student here is sufficiently trained, unlike the teacher that is early stopped. The strong regularization effect brought by an early-stopped teacher may further boost student performance.
>
>
> **___**
>
> ### ***Minor***
>
> We greatly appreciate the reviewer for pointing out these issues and we will revise accordingly per the reviewer's suggestions.
>
> __Reference__
>
> [1] Learning Student-Friendly Teacher Networks for Knowledge Distillation. Park et al., 2021.

---

> > ### Comment · Reviewer_Xgmp · 2023-12-04
> >
> > I appreciate your response and believe the promised changes will improve the paper. Although, in combination with concerns raised by other reviewers I am not convinced to improve my score.

---

### Official Review · Reviewer_peLw · 2023-10-30

**Soundness:** 4 excellent
**Presentation:** 3 good
**Contribution:** 3 good
**Rating:** 6
**Confidence:** 3

**Summary:**

This paper studies knowledge distillation and, in particular, the extent to which the teacher-model affects the student's performance.

The starting point of the authors' investigation is recent literature which suggests that the effectiveness of knowledge distillation hinges on the teacher's capability to approximate the true label distribution of training inputs. The authors show theoretical evidence which suggest that, as long as the feature extractor of the learner model is (i) Lipschitz continuous; (ii) is robust to feature transformations;  the ERM minimizer can approximate the true label distribution. Then, based on their theoretical considerations, the authors propose the "SoTeacher"  training method which incorporates Lipschitz regularization and consistency regularization into the teacher's training process.

**Strengths:**

Interesting theoretical approach for understanding the mechanics of distillation along with several experimental evidence to support the author's claim. The proposed method seems to give consistent improvements whenever applied (although sometimes rather marginal).

**Weaknesses:**

— The proposed method often times gives marginal improvements, while it introduces two extra hyper-parameters for the training of the teacher model (which is typically very costly to re-train in practical large-scale applications. That said, the authors do show that their method is relatively robust to the choice of these hyperparameters.)

— The role of the teacher as a "provider of estimates for the unknown Bayes conditional probability distribution" is a theory for why distillation works that applies well mainly in the context of multi-class classification, and especially in the case where the input is images. (Indeed, there are other explanations for why knowledge distillation works, as it can be seen as a curriculum learning mechanism, a regularization mechanism etc see e.g. [1])

In that sense, I feel that the author should either make the above more explicit in the text, i.e., explicitly restrict the scope of their claims to multi-classifcation and images, or provide evidence that their technique gives substantial improvements on binary classification tasks in NLP datasets (but even in vision datasets).

— One of the main reasons why knowledge distillation is such a popular technique, is because the teacher can generate pseudo-labels for new, unlabeled examples, increasing the size of the student's dataset. (This is known as semi-supervised distillation, or distillation with unlabeled examples, see e.g. [2, 3, 4]. ) It seems that the current approach, typically decreases the test-accuracy of the teacher model — a fact which typically impairs the student's performance in the semi-supervised setting [3, 4].


[1] Understanding and Improving Knowledge Distillation [Tang∗, Shivanna, Zhao, Lin, Singh, H.Chi, Jain]

[2] Big self-supervised models are strong semi-supervised learners [Chen, Kornblith, Swersky, Norouzi, Hinton]

[3 ]Does Knowledge Distillation Really Work? [Stanton, Izmailov,  Kirichenko, Alemi, Wilson]

[4] Weighted Distillation with Unlabeled Examples [Iliopoulos, Kontonis, Baykal, Trinh, Menghani, Vee]

**Questions:**

— Does the proposed method and theory works well/applies in NLP datasets/binary classification contexts?

---

> ### Author Response · Authors · 2023-11-23
>
> Thank you for the valuable comments and suggestions. Please see our detailed response below.
>
> __Q. Applicability of the theory__
>
> We note that our theory can be readily applied to binary classification case as binary classification can be viewed as a special case of multi-class classification. This is covered by our theory as we didn't set any assumptions on the dimension of the output prediction.
>
> We note that for NLP datasets, our theory needs to be slightly modified. Instead of assuming a "mixture of features" data model for the vision tasks, we may need a "bag of words" data model for the NLP tasks. Other theorems in our theory can be readily adapted as long as the NLP task can be formulated as a multi-class classification problem.
>
>
>
> __Q. Distillation with unlabeled data__
>
> We note that our proposed method may not necessarily hurt the distillation performance on unlabeled data. The distillation on unlabeled data often works the best if the student learns from the soft predictions of the teacher. In this case, a teacher that can approximate the true label distribution of the unlabeled dataset may be optimal, which may not necessarily coincide with a teacher with the best test accuracy. We thank the reviewer for bringing this and the relevant papers [1, 2, 3, 4] and will include this discussion in the revision.
>
> __Reference__
>
> [1] Understanding and Improving Knowledge Distillation [Tang∗, Shivanna, Zhao, Lin, Singh, H.Chi, Jain]
>
> [2] Big self-supervised models are strong semi-supervised learners [Chen, Kornblith, Swersky, Norouzi, Hinton]
>
> [3] Does Knowledge Distillation Really Work? [Stanton, Izmailov, Kirichenko, Alemi, Wilson]
>
> [4] Weighted Distillation with Unlabeled Examples [Iliopoulos, Kontonis, Baykal, Trinh, Menghani, Vee]

---

### Official Review · Reviewer_c1De · 2023-10-31

**Soundness:** 3 good
**Presentation:** 2 fair
**Contribution:** 3 good
**Rating:** 5
**Confidence:** 4

**Summary:**

This paper presents SoTeacher, a method to train teacher models that lead to better student models after knowledge distillation. The main idea is to incorporate Lipschitz and consistency regularization in training the teacher model. The authors claim that this is motivated by the theoretical analysis that the authors conduct, which shows that the ERM minimizer can approximate the true label distribution as long as the feature extractor of the teacher is Lipschitz continuous and robust to feature transformations.

**Strengths:**

* Training better-suited and student-oriented teachers is an important problem in knowledge distillation. Theoretical analysis of distillation is also of interest to the field.
* The theory sheds light on why distillation may work in practice: a teacher network can learn the true label distribution of the data under mild assumptions.
* Some empirical results are presented that support the effectiveness of the method.

**Weaknesses:**

* The dependence on $\delta$ in Lemma 3.5 seems to be very strong. Can Matrix Bernstein be used instead of Chebyshev to get a $\log(1/\delta)$ dependence?
* The proposed theory does not explain why (higher test accuracy) teachers may not lead to better students.
* The method seems to be at odds with the theory. If the assumptions of Lipschitz continuity and transformation-robustness of the feature extractor tend to be satisfied in practice, then the theoretical results hold. So, why do we need the additional regularization terms that are introduced by SoTeacher?
* The approach introduces two additional hyperparameters corresponding to the two additional regularization terms that have to be tuned.
* Evaluations on ImageNet, the largest and most real-world dataset considered, show only a marginal improvement.

**Questions:**

1. Does the presented theory explain why a better teacher may not necessarily lead to a better student?
2. Why does the method not perform as well on ImageNet?

---

> ### Author Response · Authors · 2023-11-23
>
> Thank you for the valuable comments and suggestions. Please see our detailed response below.
>
> __Q. Dependence on $\delta$ in Lemma 3.5__
>
> We thank the reviewer for the great suggestion. The dependence can indeed be improved using advanced concentration inequalities. We will modify the bound accordingly in the revision.
>
>
>
> __Q. Explanation of "teacher overfitting" by our theory__
>
> Our theory suggests that the "teacher overfitting" phenomenon, namely that a converged teacher leads to worse student performance, may result from unsatisfactory feature extractors that fail to match regularity constraints, especially the transformation-robust condition. We briefly mentioned this as a motivation for our method in Section 4. Per the reviewer's question, we realized that we may need to explain this in a standalone section. We will update accordingly in the revision.
>
>
>
> __Q. Connection between the method and the theory__
>
> We note that the regularity conditions of the feature extractors are not always satisfied in practice. As mentioned in Section 4, the Lipschitz continuity is probably satisfied in practice. However, the transformation-robustness may not be satisfied in practice, especially when the data is scarce. Therefore, it is important to explicitly enforce such regularity conditions as proposed by our method.
>
>
>
> __Q. Introduction of additional hyperparameters__
>
> We note that although our method introduced additional hyperparameters, the end performance is not sensitive to their values, as mentioned in Section 5.2.
>
>
>
> __Q. Improvement on ImageNet__
>
> We agree that on the large dataset, our method doesn't achieve significant performance improvement. This can be partly explained by our theory, as with sufficient training data, the regularity conditions, especially the transformation robustness, can already be satisfied.

---

### Official Review · Reviewer_UYE1 · 2023-11-01

**Soundness:** 3 good
**Presentation:** 3 good
**Contribution:** 2 fair
**Rating:** 5
**Confidence:** 3

**Summary:**

This paper studies the problem on how to conduct teacher training to improve student model in knowledge distillation setting. Based on recent findings that the effectiveness of knowledge distillation hinges on the teacher’s capability to approximate the true label distribution of training inputs, this paper theoretically establish that the ERM minimizer can approximate the true label distribution of training data as long as the feature extractor of the learner network is Lipschitz continuous and is robust to feature transformations. In light of the theory, the paper proposes a teacher training method called SoTeacher which incorporates Lipschitz regularization and consistency regularization into ERM. Experiments on benchmark datasets using various knowledge distillation algorithms and teacher student pairs confirm that the proposed teacher training method can improve student accuracy consistently.

**Strengths:**

Originality:
- This paper studies the feasibility of training the teacher to learn the true label distribution of its training data.
- The paper theoretically proves that when the network is Lipschitz continuous and is robust to feature transformations, the ERM minimizer can approximate the true label distribution of training data. Empirical experiments show that adding Lipschitz and Consistency regularizations can improve student accuracy consistently.

Quality:
- The paper is well-written and easy to follow.

Significance:
- Knowledge distillation is an important topic in both academia and industry. This paper studies the problem on how to conduct teacher training to better approximate the true label distribution on training data, which could benefit the community.

**Weaknesses:**

- Per my understanding, to make the teacher learn the true label distribution of its training data, the key is to resolve the overfitting issue on training data, because a teacher trained towards convergence tends to overfit the hard label. This paper proposes to add Lipschitz and Consistency regularizations to alleviate the overfitting issue. It would be great if the paper can have a more comprehensive comparison with other methods that were proposed to solve the same issue, e.g., early stopping. As shown in Figure 1(b), it seems the Standard teacher early stopped after 150 epochs has a pretty decent performance.
- The Lipschitz and Consistency regularizations are one of the most important contributions of this paper, but the readers need to read Appendix to fully understand how these two regularizations are applied. Considering the length of the Appendix is not very long, I would suggest making them clear in the main paper.
- In the experiments, it is unclear how the number of epochs impact the results.

**Questions:**

My questions are listed in the weaknesses section.

---

> ### Author Response · Authors · 2023-11-23
>
> Thank you for the valuable comments and suggestions. Please see our detailed response below.
>
> __Q. Compare with early stopping and other regularization methods__
>
> We note that properly early stopping the teacher training may indeed be able to achieve student performance comparable to SoTeacher. However, as also mentioned in Line 356 in our paper, it is often difficult to locate the right epoch number for early stopping. Slight deviation from the best epoch number may significantly impair the student's performance. A meticulous search for the best epoch number will induce high computation costs as one may need to **conduct full student training for each epoch number**. In contrast, SoTeacher can achieve the best student performance even when the teacher fully converges, which will be much more efficient and friendly to use in practice.
>
> For other popular training or post-training techniques including classic regularizations such as l1 and l2, label smoothing, advanced data augmentation, temperature scaling, and uncertainty learning methods, we have provided a detailed comparison in Appendix F.
>
> __Q. Location of the details of the proposed methods__
>
> Thanks for your suggestion. We will move the details of the proposed methods into the main paper.
>
>
>
> __Q. Effect of the number of training epochs__
>
> In this paper, we follow the widely used settings for training the teacher, namely 240 epochs for CIFAR-100 and 90 epochs for ImageNet and Tiny-ImageNet. We will explore alternative training settings in the revision.

---

### Meta-Review · Area_Chair_SZhW · 2023-12-05

**Metareview:**

The authors build on the recent theoretical results that the effectiveness of the teacher deteriorates when it cannot correctly predict the label distribution. The authors claim that Lipschitz bounded robust to feature extractors are better suited for this task and propose a Lipschitz regularizer to improve distillation.
- There seems to be a misalignment between the theoretical results and our intuitive understanding of distillation. The main theoretical claim is that Lipschitz bounded feature extractors better approximate the true label distribution. Thus, addressing this issue is both relevant to noise robustness (since normally the teacher would overfit to the training labels) or a multi-label setting where there is no clear true label. It is unclear why a teacher with a better generalization may not necessarily translate to a better student.
- The empirical improvements are marginal. Also, the authors do not compare to simpler methods such as early stopping.

**Justification For Why Not Higher Score:**

I am suggesting accepting (poster) since the overall response by the reviewers is positive, but there are still some concerns that have not been fully addressed.

**Justification For Why Not Lower Score:**

There are some benefits in the theoretical findings of the paper that might also be appealing to practitioners from an applied standpoint.

---

### Decision · Program_Chairs · 2024-01-16

Accept (poster)